# ST-HHOL: Spatio-Temporal Hierarchical Hypergraph Online Learning for Crime Prediction

**Keqing Du**[1], **Yufan Kang**[2], **Xinyu Yang**[1]\*, **Wei Shao**[3,4]\*
[1]Xi'an Jiaotong University, [2]Monash University, [3]Data61, CSIRO, [4]University of New South Wales
dukeqing@stu.xjtu.edu.cn, tina.kang@monash.edu,
yxyphd@mail.xjtu.edu.cn, wei.shao@{data61.csiro.au, unsw.edu.au}

## Abstract

Crime prediction is a critical yet challenging task in urban spatio-temporal forecasting. Sparse crime records alone are insufficient to capture latent high-order patterns shaped by heterogeneous contextual factors with spatial and criminal specificity, while high non-stationarity renders conventional offline models ineffective against concept drift. To tackle these challenges, we propose a Spatio-Temporal Hierarchical Hypergraph Online Learning framework named ST-HHOL. First, we propose a hierarchical hypergraph convolution network that integrates crime data with heterogeneous contextual factors to uncover dual-specific crime patterns and their co-occurrence relations. Second, we introduce an iterative online learning strategy to address concept drift by employing frequent fine-tuning for short-term dynamics and periodic retraining for long-term shifts. Moreover, we adopt a Partially-Frozen LLM that leverages pre-trained sequence priors while adapting its attention mechanisms to crime-specific dependencies, enhancing spatio-temporal reasoning under sparse supervision. Extensive experiments on four real-world datasets demonstrate that ST-HHOL consistently outperforms state-of-the-art methods in terms of accuracy and robustness, while also providing enhanced interpretability. Code is available at https://github.com/777Rebecca/ST-HHOL.

## 1 Introduction

Crime prediction is a critical task in urban spatio-temporal forecasting, with significant implications for public safety and social stability. Recent advances in deep neural networks have significantly promoted the development of this field, leveraging attention mechanisms (Xia et al., 2021; Rayhan & Hashem, 2023) to model dynamic crime correlations, and graph neural networks (GNNs) (Zheng et al., 2020; Wu et al., 2020a; Zhou et al., 2024) to capture temporal evolution and spatial heterogeneity.

However, sparse crime data alone cannot reveal the multifaceted crime patterns with *spatial* and *criminal specificity*. Potential risk depends on the joint influence of spatio-temporal factors, such as environment, mobility, and weather, whose type and strength vary across regions and crime categories. For example, as shown in Figure 1(a), assaults often rise around bars during late hours, whereas thefts are more prevalent near subway stations during daytime. Prior studies that incorporate auxiliary data typically model it as homogeneous graphs, pairwise graphs, or rely on simple feature fusion (Huang et al., 2018; Zhou et al., 2024; Zhu et al., 2022), which cannot capture the high-order, dual-specific interactions of coexisting factors in shaping crime occurrence. Consequently, these methods fail to account for the latent heterogeneity of crime patterns and lead to biased predictions.

Moreover, crime data exhibits pronounced *non-stationarity*, challenging conventional offline models that struggle to adapt to concept drift (Tsymbal, 2004). As illustrated in Figure 1(b), crime counts in different areas fluctuate sharply over a few days, particularly in regions outlined by the white border. Seasonal and environmental factors further intensify these dynamics, leading to distributional shifts where $P_{\text{train}}(\mathbf{Y}|\mathbf{X}) \neq P_{\text{test}}(\mathbf{Y}|\mathbf{X})$. Although recent approaches (Xia et al., 2023; Zheng et al., 2022;

---

\*Corresponding author.

Yang et al., 2022) attempt to extract invariant terms, they often assume complete data availability and static inter-variable relationships, limiting their adaptability to emerging dynamics.

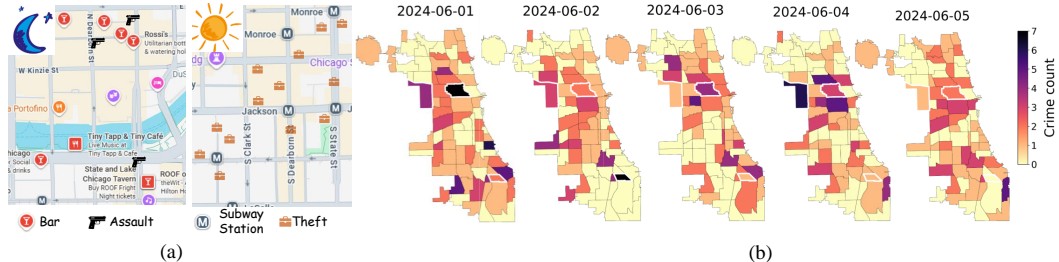

Figure 1: (a) Example of the heterogeneity of latent crime patterns: assaults near nightlife at midnight, thefts near subways in daytime. (b) Spatial distribution of "Criminal Damage" in Chicago (June 1–5, 2024), with the white-bordered region indicating non-stationarity.

To tackle these challenges, we propose **ST-HHOL**, a novel **S**patio-**T**emporal **H**ierarchical **H**ypergraph **O**nline **L**earning framework for crime prediction. To uncover latent crime patterns, we propose hierarchical hypergraph convolution network (HHGCN), a novel idea that first builds heterogeneous hypergraphs to model latent crime patterns by integrating crime data with specific contextual factors, and then constructs homogeneous hypergraphs to model crime co-occurrence relations. HHGCN jointly captures the dual-specific crime patterns and their interaction, yielding richer and more interpretable representations. To handle concept drift in non-stationary crime data, ST-HHOL adopts an iterative online learning strategy that separates short-term fluctuations from long-term gradual shifts in streaming data. It leverages frequent fine-tuning to adapt to rapid temporal dynamics, while periodic retraining addresses long-term spatial shifts. Moreover, to address the limited supervision of sparse crime records, we incorporate a Partially-Frozen LLM (PF-LLM) as a spatio-temporal dependency learner. PF-LLM leverages the pre-trained sequence modeling priors of GPT-2, while selectively adapting its attention mechanism to capture crime-specific dependencies.

Our main contributions are summarized as follows:

- We propose ST-HHOL, an online spatio-temporal crime prediction framework that jointly captures high-order, dual-specific crime patterns and adapts to concept drift.
- We develop an iterative online learning strategy that combines frequent fine-tuning for short-term fluctuations with periodic retraining for long-term concept drifts.
- We design a hierarchical hypergraph convolution network that integrates contextual factors to uncover latent crime patterns with spatial and criminal specificity, and their co-occurrence relations.
- We conduct extensive experiments on four real-world urban crime datasets, showing that ST-HHOL consistently surpasses state-of-the-art baselines in both accuracy and robustness, while offering improved interpretability.

## 2 RELATED WORK

**Crime Prediction.** Beyond conventional statistical and machine learning approaches (Catlett et al., 2018; Kumar et al., 2020), GNN-based models—such as STGCN (Yu et al., 2018), DCRNN (Li et al., 2017), AGCRN (Bai et al., 2020), MTGNN (Wu et al., 2020b), and GMAN (Zheng et al., 2020)—exhibit considerable promise for capturing intricate spatial and temporal dependencies. To further enhance predictive performance, some studies have integrated auxiliary data sources, including points of interest (POI), 311 service requests (Yang et al., 2018; Wang et al., 2016; Huang et al., 2018), and mobility data (e.g., taxi and bike inflow and outflow) (Zhao et al., 2022; Wu et al., 2020a), either fusing them with crime records or constructing multi-graph structures to enrich representations.

Nevertheless, crime records are sparse and have a skewed distribution, compounded by pronounced heterogeneity across crime types. To tackle these challenges, prior works introduce a tensor decomposition framework (Zhao & Tang, 2017a) to capture inter-regional and inter-type dependencies, or employ transfer learning (Zhao & Tang, 2017b) to adapt knowledge from data-rich to data-scarce areas. Furthermore, fine-grained spatial partitioning further amplifies the imbalance and risk of overfitting. ST-Trans (Wu et al., 2020a) addresses this by jointly modeling spatial, temporal, and semantic

dependencies with adversarial training to enhance rare-event prediction. Similarly, NAHC (Liang et al., 2022) combines multi-graph convolution with attention mechanisms to improve hour-level forecasting accuracy.

Recognizing that spatial correlations in crime data often exhibit high-order dependencies, recent efforts have turned to hypergraph-based methods. ST-SHN (Xia et al., 2021) demonstrates the utility of hypergraphs in capturing such cross-regional interactions, extending spatial correlations beyond conventional N-hop neighborhoods. ST-HSL (Li et al., 2022b) incorporates a self-supervised framework, aligning global hypergraph-based features with local CNN-based ones to strengthen model robustness. Building on this, HCL (Liang et al., 2024) proposes Hawkes-enhanced temporal augmentation and a negative-free contrastive strategy to identify co-occurrence patterns. Most recently, ST-MoGE (Wu et al., 2024) adopts a Mixture-of-Graph-Experts model to jointly address spatial and semantic heterogeneity in crime.

Despite these advances, most prior work remains offline and cannot adapt to pronounced non-stationarity or concept drift. Existing hypergraph-based crime models also build homogeneous or flat hierarchical hypergraphs directly from sparse crime records, overlooking high-order interactions between heterogeneous contextual factors and crime semantics—resulting in mixed semantics and unstable patterns under sparsity. In contrast, our approach (1) models heterogeneous latent crime patterns by assigning factor-specific hyperedges to each region–crime anchor and capturing their dynamic coupling, and (2) introduces hierarchical temporal adaptation to address multi-scale concept drift—two abilities absent in prior hypergraph-based methods.

## 3 PRELIMINARIES

In this section, we first introduce the relevant definitions and formulate the research problem. The raw crime records provide timestamps, location coordinates, crime types, and additional descriptions. We organize them into processed crime data $\mathbf{X} \in \mathbb{R}^{N \times T \times C}$, where $N$, $T$, and $C$ denote the number of regions, time slots, and main crime types, respectively. Each entry $x_{n,c}^t$ represents the occurrence of crime type $c$ in region $n$ during time slot $t$. Since crime data are often sparse and in skewed distribution, zero observations do not indicate zero risk, we further incorporate auxiliary urban data—such as 311 service requests, weather conditions, and POI density—denoted as $\mathbf{S} \in \mathbb{R}^{N \times T \times M}$, where $M$ is the number of auxiliary variables. The main notations and definitions are also provided in Appendix A.

**Problem Statement.** Given the historical crime data $\mathbf{X} \in \mathbb{R}^{N \times T \times C}$ and the multi-source auxiliary data $\mathbf{S} \in \mathbb{R}^{N \times T \times M}$, the crime prediction task aims to learn a mapping function $\mathcal{F}(\cdot) : \mathbb{R}^{T \times N \times C} \rightarrow \mathbb{R}^{N \times C}$ that forecasts crime occurrences $\mathbf{X}^{T+1} \in \mathbb{R}^{N \times C}$ during the next time slot:

$$\{\mathbf{X}^{1:T}, \mathbf{S}^{1:T}\} \xrightarrow{\mathcal{F}(\cdot)} \mathbf{X}^{T+1}. \tag{1}$$

**Heterogeneous and Homogeneous Hypergraph.** A hypergraph is defined as $\mathcal{G} = \{\mathcal{V}, \mathcal{E}, \mathcal{T}_v, \mathcal{T}_e\}$, where $\mathcal{V}$ and $\mathcal{E}$ denote the sets of vertices and hyperedges, and $\mathcal{T}_v$ and $\mathcal{T}_e$ represent their respective types. If $|\mathcal{T}_v| + |\mathcal{T}_e| > 2$, the hypergraph is considered heterogeneous; otherwise, it is homogeneous. Unlike traditional pairwise graphs, where each edge connects exactly two vertices, a hyperedge can simultaneously connect multiple vertices, enabling the modeling of higher-order relations. Details of hypergraph convolution are provided in Appendix C.

## 4 METHODOLOGY

As shown in Figure 2, ST-HHOL follows an online learning paradigm to tackle crime pattern specificity and concept drift. It consists of two components: (1) a hierarchical hypergraph convolution network (HHGCN) that captures crime patterns with spatial and criminal specificity, as well as their co-occurrence relations, and (2) a spatio-temporal dependency learner that captures temporal dynamics and spatial correlations to forecast. To handle non-stationary streaming data, ST-HHOL adopts an iterative online learning strategy after warming up, where partial fine-tuning adapts to short-term fluctuations and periodic retraining addresses long-term distributional shifts. The detailed design of the pipeline and its components will be introduced in the following parts.

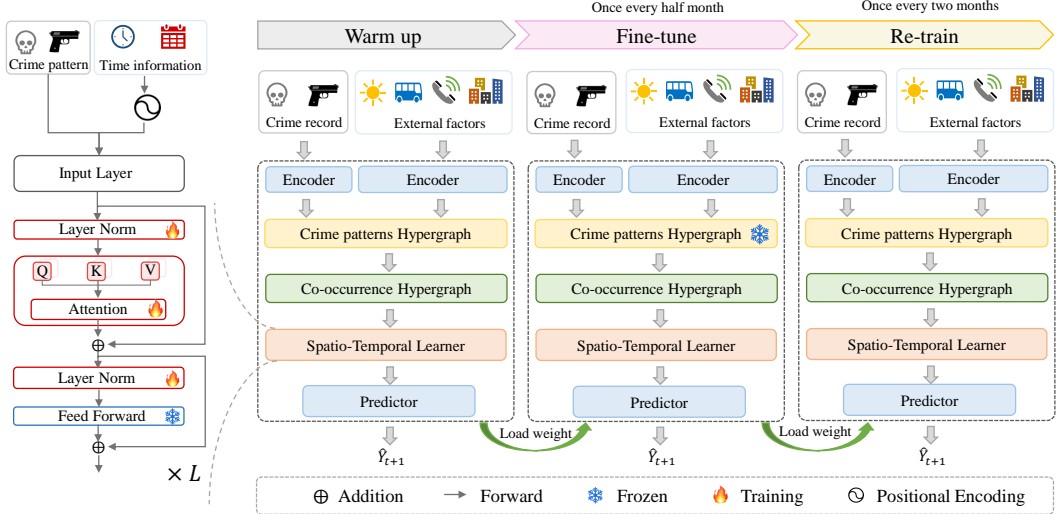

Figure 2: The framework of ST-HHOL.

## 4.1 HIERARCHICAL HYPERGRAPH CONVOLUTION NETWORK

To capture the spatial and criminal specificity of crime patterns, we propose a hierarchical hypergraph convolution network to more accurately perceive the potential risks of crimes. It first constructs heterogeneous hypergraphs $\mathcal{G}^e$ that fuse multi-source contextual signals such as POIs, 311 service requests, weather, with crime records to reveal the heterogeneous latent patterns that cannot be extracted from sparse raw counts. We then build homogeneous hypergraphs $\mathcal{G}^o$ to model the dynamic co-occurrence among latent crime patterns, as illustrated in Figure 3.

**Heterogeneous Hypergraph Construction.** Given $C$ crime types and $M$ multi-source factors, we obtain the initial embeddings $\mathbf{X}^t \in \mathbb{R}^{N \times C}$ and $\mathbf{S}^t \in \mathbb{R}^{N \times M}$ for all regions at time $t$. We construct the heterogeneous hypergraph $\mathcal{G}_t^e = (\mathcal{V}_t, \mathcal{E}_t)$, where the vertex set consists of all crime and contextual nodes: $\mathcal{V}_t = \{x_{n,c}^t\}_{n=1,c=1}^{N,C} \cup \{s_{n,m}^t\}_{n=1,m=1}^{N,M}$. Each crime embedding $x_{n,c}^t$ serves as a primary node, and together with its contextual nodes $\{s_{n,m}^t\}_{m=1}^{M}$, forms a heterogeneous hyperedge. The latent crime pattern representations are computed as:

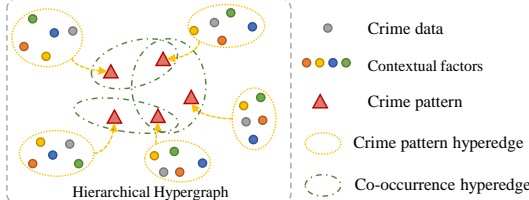

Figure 3: The hierarchical hypergraph consists of heterogeneous hypergraphs capturing crime patterns and homogeneous hypergraphs modeling co-occurrence relationships.

$$\tilde{\mathbf{X}}^t = f\big(\sigma\big(\mathbf{\Theta}_e^t \left[\mathbf{X}^t \,\|\, \mathbf{S}^t\right]\big)\big), \tag{2}$$

where $\tilde{\mathbf{X}}^t \in \mathbb{R}^{N \times C}$ denotes the latent crime patterns at time $t$, $\mathbf{\Theta}_e^t \in \mathbb{R}^{H_e \times |\mathcal{V}_t|}$ is a learnable incidence matrix mapping vertices to hyperedges, and $H_e = N \times C$ is the number of heterogeneous hyperedges. The operator $\|$ denotes feature concatenation, $\sigma(\cdot)$ is the sigmoid activation function, and $f(\cdot)$ is a nonlinear transformation implemented by a three-layer MLP with LeakyReLU activation. For any vertex $v_i^t \in \mathcal{V}_t$, its contribution to crime pattern $\tilde{x}_{n,c}^t \in \tilde{\mathbf{X}}^t$ is defined as:

$$\mathbf{\Theta}_e^t[\tilde{x}_{n,c}^t, v_i^t] = \begin{cases} 1, & v_i^t = x_{n,c}^t, \\ p_{i,n,c}^t, & v_i^t \in \{s_{n,m}^t\}_{m=1}^M, \\ 0, & \text{otherwise,} \end{cases} \tag{3}$$

where $p_{i,n,c}^t \in (0,1)$ quantifies the association strength between contextual factor $s_{n,i}^t$ and latent crime pattern $\tilde{x}_{n,c}^t$.

**Homogeneous Hypergraph Modeling.** To capture high-order spatial co-occurrence relationships, we further construct the homogeneous hypergraph $\mathcal{G}_t^o$ over the latent crime patterns. The hypergraph convolutional network aggregates the co-occurring patterns as:

$$\mathbf{E}^t = \sigma\left((\widetilde{\mathbf{D}}_R^t)^{-\frac{1}{2}}\mathbf{\Phi}^T(\widetilde{\mathbf{D}}_E^t)^{-\frac{1}{2}}\,\sigma\left((\widetilde{\mathbf{D}}_E^t)^{-\frac{1}{2}}\mathbf{\Phi}(\widetilde{\mathbf{D}}_R^t)^{-\frac{1}{2}}\tilde{\mathbf{X}}^t\right)\right), \tag{4}$$

where $\mathbf{\Phi}$ is the homogeneous incidence matrix, and $\widetilde{\mathbf{D}}_R^t$ and $\widetilde{\mathbf{D}}_E^t$ denote the degree matrices of hyperedges and vertices, respectively. For model simplification, we approximate the matrix $(\widetilde{\mathbf{D}}_R^t)^{-\frac{1}{2}}\mathbf{\Phi}^T(\widetilde{\mathbf{D}}_E^t)^{-\frac{1}{2}}$ with a learnable parameter matrix $\mathbf{\Theta}_o^t \in \mathbb{R}^{H_o \times (NC)}$, where $H_o$ is a hyperparameter denoting the number of homogeneous hyperedges.

## 4.2 SPATIO-TEMPORAL DEPENDENCY LEARNER

To enhance spatio-temporal reasoning over non-stationary and heterogeneous crime patterns, we adopt a Partially-Frozen Large Language Model (PF-LLM) (Liu et al., 2025a) built on GPT-2 (Radford et al., 2019). The key intuition is that transformer self-attention is modality-agnostic, and the pretrained feed-forward networks (FFNs) encode generalizable sequence priors and few-shot reasoning capabilities learned from large-scale text corpora. These priors remain useful even when crime data are sparse and noisy. PF-LLM freezes the FFNs to preserve these transferable reasoning abilities, while fine-tuning the attention and normalization layers to adapt them to crime-specific, spatio-temporal structures and their non-stationary dynamics.

For tokenization, we treat each region as a token and transform crime patterns $\mathbf{E} \in \mathbb{R}^{N \times T \times C}$ into a sequence $\mathbf{E} = \{\mathbf{E}_1, \ldots, \mathbf{E}_N\}$, where $\mathbf{E}_n \in \mathbb{R}^{T \times C}$ encodes the $C$ type of crime patterns in region $n$ over $T$ time steps. To preserve temporal semantics, we extract day-of-week ($t_d$) and month-of-year ($t_m$) indicators from each timestamp and encode them using one-hot vectors followed by sinusoidal positional encoding $\sin(\cdot)$. The resulting temporal embedding $\mathbf{E}_T$ combines multiple periodic components $\mathbf{E}_T^d$ and $\mathbf{E}_T^m$:

$$\mathbf{E}_T^d = \sin(t_d), \qquad \mathbf{E}_T^m = \sin(t_m), \qquad \mathbf{E}_T = \mathbf{E}_T^d + \mathbf{E}_T^m. \tag{5}$$

To align the input dimension with the hidden space of GPT-2, we apply a set of non-linear transformation layers—one for each crime type and an additional one for temporal features—to project the raw inputs into the model's latent space. We also incorporate a Pre-Layer Normalization (Pre-LN) scheme (Huang et al., 2023; Liu et al., 2025b) to stabilize training and accelerate convergence. The resulting representations are then fed into the partially frozen GPT-2 architecture as follows:

$$\begin{aligned}
\mathbf{H}^1 &= \text{Concat}(f_1(\mathbf{E}_1), \ldots, f_C(\mathbf{E}_C), f_{C+1}(\mathbf{E}_T)), \\
\bar{\mathbf{H}}^l &= \text{MHA}\big(\text{LN}\big(\mathbf{H}^l\big)\big) + \mathbf{H}^l, \qquad \mathbf{H}^{l+1} = \text{FFN}\big(\text{LN}\big(\bar{\mathbf{H}}^l\big)\big) + \bar{\mathbf{H}}^l,
\end{aligned} \tag{6}$$

where $f_1, \ldots, f_C, f_{C+1}$ are non-linear functions that project $C$ crime types and one temporal feature into the model's latent space. $\{\mathbf{E}_1, \ldots, \mathbf{E}_C\} \in \mathbb{R}^{C \times N \times T}$ represent the tokens distinguished by crime types. $\text{Concat}(\cdot)$ denotes the concatenation operation. $\mathbf{H}^1$ denotes the initial input embedding, $\mathbf{H}^l$ is the input to the $l$-th layer, $\bar{\mathbf{H}}^l$ is the intermediate representation produced by the unfrozen Layer Normalization (LN) and Multi-Head Attention (MHA) components, and $\mathbf{H}^{l+1}$ is the output by applying the unfrozen LN and the frozen FFN. The MHA, LN, FFN operation is formally defined as:

$$\text{MHA}(\mathbf{H}^l) = \text{Concat}(\text{head}_1, \ldots, \text{head}_h)\mathbf{W}^O, \qquad \text{LN}(\mathbf{H}^l) = \boldsymbol{\gamma} \odot \frac{\mathbf{H}^l - \boldsymbol{\mu}}{\boldsymbol{\sigma}} + \boldsymbol{\beta}, \tag{7}$$

$$\text{head}_i = \text{softmax}\left(\frac{(\mathbf{H}^l\mathbf{W}^Q)(\mathbf{H}^l\mathbf{W}^K)^T}{\sqrt{d_k}}\right)\mathbf{H}^l\mathbf{W}^V, \; \text{FFN}(\bar{\mathbf{H}}^l) = \text{ReLU}(\mathbf{W}_1\bar{\mathbf{H}}^l + \mathbf{b}_1)\mathbf{W}_2 + \mathbf{b}_2,$$

where $\boldsymbol{\gamma}$ and $\boldsymbol{\beta}$ are learnable scaling and translation parameters. $\boldsymbol{\mu}$ and $\boldsymbol{\sigma}$ represent the mean and standard deviation, respectively. $\odot$ denotes the Hadamard product. $\mathbf{W}^O$, $\mathbf{W}^Q$, $\mathbf{W}^K$, and $\mathbf{W}^V$ denote the projection mappings for the output, query, key, and value, respectively. $\text{head}_i$ represents the $i$-th attention head, $d_k$ represents the dimension of each attention head. $\mathbf{W}_1$ and $\mathbf{W}_2$ are the weights of the linear transformations, and $\mathbf{b}_1, \mathbf{b}_2$ are bias terms.

## 4.3 ONLINE LEARNING STRATEGY

Conventional offline models often implicitly assume a stationary data distribution. However, crime patterns are inherently susceptible to *concept drift*, which can be formally characterized as:

$$\exists \, \tau > 0, \quad \mathrm{D}_{\mathrm{KL}}(P_t(\mathbf{Y} \mid \mathbf{X}) \, \| \, P_{t+\tau}(\mathbf{Y} \mid \mathbf{X})) \geq \delta, \tag{8}$$

where $P_t(\mathbf{Y} \mid \mathbf{X})$ denotes the conditional distribution of crime events at time $t$. $\mathrm{D}_{\mathrm{KL}}$ denotes the Kullback–Leibler divergence, and $\delta$ is a predefined divergence threshold.

To address the non-stationary nature of crime dynamics, we propose an iterative online learning strategy for ST-HHOL that explicitly disentangles spatially invariant and temporally variant components. While crime co-occurrences exhibit volatility over time, as shown in Figure 1(b), the heterogeneous components driving crime patterns and their strengths remain relatively stable across space. We parameterize short-term fluctuations and long-term gradual shifts as $\{\mathbf{\Theta}_s, \mathbf{\Theta}_d(t)\} \subset \mathbf{\Theta}$, where:

- $\mathbf{\Theta}_s$**: Parameters encoding spatial invariance and long-term gradual shifts.** These parameters are associated with the first-stage hypergraph $\mathcal{G}_t^e$, which constructs crime patterns from multi-source inputs $\mathbf{S}^t$ and crime records $\mathbf{X}^t$. Given their dependence on relatively stationary spatial attributes, such as socioeconomic indicators and POIs, $\mathbf{\Theta}_s$ evolves slowly over time and can be modeled as:

$$\frac{d\mathbf{\Theta}_s}{dt} \approx 0, \qquad \mathbf{\Theta}_s^{t+1} = \mathbf{\Theta}_s^t + \varepsilon_t, \qquad \varepsilon_t \sim \mathcal{N}(0, \sigma^2), \quad \sigma^2 \ll 1. \tag{9}$$

- $\mathbf{\Theta}_d(t)$**: Parameters capturing short-term temporal fluctuations.** These parameters are associated with the second-stage hypergraph $\mathcal{G}_t^o$, which captures rapidly evolving co-occurrence structures among crime patterns $\mathbf{E}^t$. Unlike $\mathbf{\Theta}_s$, $\mathbf{\Theta}_d(t)$ reacts to abrupt regional fluctuations, characterized by $\|\Delta\mathbf{E}_i^t\|_1 \gg 0$. Such shocks trigger enforcement adjustment $L_i^{t+1} = L_i^t + \psi(\|\Delta\mathbf{E}_i^t\|_1)$ and consequently reshape regional interactions as $D_{ij}^{t+1} = D_{ij}^t + g(L_i^{t+1}, L_j^{t+1})$, where $\psi$ and $g$ denote the response and coupling mechanisms, respectively. Consequently, the dynamic parameters evolve more rapidly and can be modeled as:

$$\mathbf{\Theta}_d^{t+1} = \mathbf{\Theta}_d^t + \eta_t, \qquad \eta_t \sim \mathcal{N}(0, \sigma_t^2), \qquad \sigma_t^2 \gg \sigma^2. \tag{10}$$

Thus, after an initial warm-up, ST-HHOL adopts an iterative two-phase update scheme. This design balances short-term responsiveness with long-term stability and improves robustness under non-stationary crime dynamics:

- **Fine-tuning Phase:** Every $\tau$ steps, the spatially invariant parameters $\mathbf{\Theta}_s$ are frozen and only the temporally dynamic parameters $\mathbf{\Theta}_d(t)$, along with the spatio-temporal dependency learner $\mathbf{\Theta}_{\mathrm{PF\text{-}LLM}}$, are updated. This phase enables rapid adaptation to recent fluctuations by solving: $\min_{\mathbf{\Theta} \setminus \mathbf{\Theta}_s} \mathcal{L}\big(\mathbf{Y}^{t+1}, \hat{\mathbf{Y}}^{t+1}(\mathbf{\Theta} \setminus \mathbf{\Theta}_s)\big);$

- **Retraining Phase:** Every $T$ steps ($T > \tau$), $\mathbf{\Theta}_s$ are unfrozen and jointly updated with $\mathbf{\Theta}_d(t)$. This phase captures long-term gradual shifts and evolving co-occurrence structures by optimizing: $\min_{\mathbf{\Theta}} \mathcal{L}\big(\mathbf{Y}^{t+1}, \hat{\mathbf{Y}}^{t+1}(\mathbf{\Theta})\big).$

## 4.4 MODEL PREDICTION AND OPTIMIZATION

Under different online learning modes, ST-HHOL maintains a unified prediction and loss computation framework. The output $\mathbf{H}^{l+1}$ from the final layer of the spatio-temporal dependency learner is passed through crime-specific regression heads to generate future forecasts:

$$\hat{\mathbf{Y}}^{t+1} = \mathrm{Concat}\big(\mathrm{RConv}_1(\mathbf{H}_1^{l+1}), \ldots, \mathrm{RConv}_c(\mathbf{H}_c^{l+1})\big), \tag{11}$$

where $\hat{\mathbf{Y}}^{t+1} \in \mathbb{R}^{N \times C}$ denotes the predicted crime records for the next time slot.

So far, the loss function of ST-HHOL consists of prediction loss and hypergraph regularization loss:

$$\mathcal{L} = \left\|\mathbf{Y}^{t+1} - \hat{\mathbf{Y}}^{t+1}\right\|_2^2 + \lambda_1 \left\|\mathbf{\Theta}_e^{t+1}\right\|_2^2 + \lambda_2 \left\|\mathbf{\Theta}_o^{t+1}\right\|_2^2, \tag{12}$$

where $\mathbf{Y}^{t+1}$ is the ground-truth value at time slot $t + 1$, $\mathbf{\Theta}_e^{t+1}$ and $\mathbf{\Theta}_o^{t+1}$ are the incident matrix elements of the crime patterns within heterogeneous hypergraph $\mathcal{G}_{t+1}^e$ and the co-occurrence relationships within homogeneous hypergraph $\mathcal{G}_{t+1}^o$ at $t + 1$, respectively. $\lambda_1$ and $\lambda_2$ are two hyperparameters balancing the loss terms.

## 5 EXPERIMENTS

In this section, we evaluate our proposed ST-HHOL framework on four real-world urban crime datasets. Extensive experiments are designed to address the following research questions:

- **RQ1:** How does ST-HHOL perform on crime quantity and occurrence prediction?
- **RQ2:** How does each module within ST-HHOL enhance the overall model performance?
- **RQ3:** How do varying retraining and fine-tuning frequencies affect the performance?
- **RQ4:** What impact do different hyperparameter settings have on ST-HHOL?
- **RQ5:** Does the hierarchical hypergraph constructed in ST-HHOL enhance model interpretability?

Table 1: Summary of the four urban crime datasets.

| Datasets | Region Num | Time Span | Time Interval | Crime Data |
|---|---|---|---|---|
| CHI | 77 | January 1, 2023 – December 31, 2024 | 1 day | Theft, Battery, Assault, Damage |
| NYC | 123 | January 1, 2022 – December 31, 2024 | 1 day | Larceny, Assault, Mischief, Robbery |
| PHI | 6 | January 1, 2023 – December 31, 2024 | 1 day | Theft, Assault, Vehicle, Mischief |
| TOR | 158 | January 1, 2023 – December 31, 2024 | 1 day | Assault, B&E, Robbery, Theft |

**Datasets & Baselines.** We conduct experiments on four urban crime datasets from Chicago (CHI), New York City (NYC), Philadelphia (PHI), and Toronto (TOR) with multi-source contextual data, including 311 service requests, weather, and POI distributions. For comparison, we include a broad range of baselines: statistical methods (SVM (1998), ARIMA (2015)), spatio-temporal forecasting models (DCRNN (2017), STGCN (2018), AGCRN (2020), MTGNN (2020b), GMAN (2020), MoSSL (2024)), crime prediction models (DeepCrime (2018), ST-HSL (2022b), ST-SHN (2021)), and online learning models (DLF (2024), FSNet (2023), OneNet (2023)). More details about the datasets and baselines are provided in Appendix D. We evaluate offline baselines under two settings (*Pre-trained* and *Re-trained*), while online baselines under their inherent *Online* setting:

- **Pre-trained**: models are trained once on the full dataset under the conventional offline setting.
- **Re-trained**: offline models are adapted into an online variant, which is periodically retrained every two months using streaming data after an initial warm-up.
- **Online**: online models incrementally update according to their native online learning strategies using streaming data.

**Implementation Details.** All models are implemented in PyTorch 2.0 and trained on an NVIDIA RTX 3090 GPU. We use the Adam optimizer with a batch size of 32, an initial learning rate of $1e^{-3}$, and a decay factor of $1e^{-4}$. The number of hyperedges in $\mathcal{G}^o$ is set to 64. PF-LLM utilizes GPT-2 (Small), comprising 2 layers, 12 attention heads, and a hidden size of 768. $\lambda_1$ and $\lambda_2$ are both set to 0.1. The dataset is chronologically split into a warm-up and online training phase in a 25:75 ratio. Following (Liang et al., 2024; Wu et al., 2024), we set the temporal input length to 7 and the forecasting horizon to 1. More settings and evaluation metrics are provided in Appendix D.3-D.4.

Table 2: Overall performance of crime quantity prediction over CHI, NYC, PHI, and TOR datasets. The results are 5-run error comparison, the **bold** font means the best result.

| | Method | CHI | | | | NYC | | | | PHI | | | | TOR | | | |
|---|---|---|---|---|---|---|---|---|---|---|---|---|---|---|---|---|---|
| | | Theft | | Battery | | Larceny | | Assault | | Vehicle | | Mischief | | Assault | | B&E | |
| | | MAE | MAPE | MAE | MAPE | MAE | MAPE | MAE | MAPE | MAE | MAPE | MAE | MAPE | MAE | MAPE | MAE | MAPE |
| *Pre-trained* | SVM | 1.54 | 0.51 | 0.94 | 0.56 | 1.81 | 0.61 | 0.79 | 0.53 | 2.85 | 0.49 | 2.95 | 0.48 | 0.74 | 0.53 | 1.62 | 0.56 |
| | ARIMA | 1.52 | 0.52 | 0.99 | 0.58 | 1.17 | 0.63 | 0.75 | 0.50 | 2.62 | 0.49 | 3.34 | 0.58 | 0.71 | 0.50 | 1.56 | 0.55 |
| | DCRNN | 3.04±.30 | 0.57±.06 | 1.94±.13 | 0.73±.01 | 1.56±.09 | 0.54±.03 | 0.92±.01 | 0.52±.03 | 3.23±.35 | 0.53±.02 | 3.59±.75 | 0.67±.02 | 0.79±.06 | 0.49±.04 | 1.65±.12 | 0.50±.06 |
| | STGCN | 2.92±.56 | 0.66±.11 | 2.14±.05 | 0.69±.03 | 1.55±.03 | 0.47±.04 | 0.93±.01 | 0.50±.05 | 3.64±1.66 | 0.42±.07 | 2.87±1.17 | 0.52±.05 | 0.88±.03 | 0.54±.05 | 1.58±.08 | 0.48±.07 |
| | AGCRN | 1.06±.12 | 0.54±.08 | 0.92±.15 | 0.55±.03 | 1.38±.34 | 0.78±.00 | 0.82±.04 | 0.40±.25 | 2.81±.03 | 0.51±.01 | 2.74±.24 | 0.52±.03 | 0.67±.02 | 0.40±.02 | 1.13±.12 | 0.48±.06 |
| | ST-SHN | 1.14±.09 | 0.74±.07 | 0.98±.03 | 0.79±.02 | 1.41±.05 | 0.56±.09 | 0.79±.02 | 0.38±.06 | 2.93±.47 | 0.59±.20 | 2.95±.61 | 0.50±.15 | 0.77±.04 | 0.40±.07 | 1.29±.03 | 0.52±.09 |
| | ST-HSL | 1.17±.03 | 0.55±.03 | 0.97±.02 | 0.55±.04 | 1.34±.15 | 0.50±.11 | 0.82±.15 | 0.47±.05 | 2.71±1.95 | 0.55±.13 | 2.54±.46 | 0.52±.20 | 0.69±.12 | 0.47±.07 | 1.23±.17 | 0.50±.04 |
| *Re-trained* | DCRNN | 2.91±.25 | 0.49±.05 | 1.82±.15 | 0.64±.05 | 1.42±.07 | 0.44±.03 | 0.85±.01 | 0.50±.04 | 3.09±.49 | 0.47±.01 | 3.43±.71 | 0.58±.01 | 0.74±.05 | 0.44±.03 | 1.50±.10 | 0.45±.05 |
| | STGCN | 2.82±.58 | 0.58±.09 | 2.03±.04 | 0.61±.03 | 1.41±.03 | 0.39±.04 | 0.86±.06 | 0.46±.07 | 3.54±1.60 | 0.38±.05 | 2.78±1.15 | 0.48±.05 | 0.83±.02 | 0.49±.04 | 1.43±.06 | 0.43±.06 |
| | AGCRN | 0.99±.11 | 0.50±.07 | 0.88±.17 | 0.51±.02 | 1.26±.39 | 0.74±.01 | 0.70±.01 | 0.30±.26 | 2.68±.03 | 0.41±.01 | 2.70±.22 | 0.49±.02 | 0.62±.01 | 0.35±.01 | 0.98±.10 | 0.43±.05 |
| | MTGNN | 1.45±.34 | 0.58±.18 | 1.06±.33 | 0.60±.08 | 1.44±.22 | 0.57±.05 | 1.09±.12 | 0.60±.14 | 2.59±.83 | 0.48±.12 | 2.93±.06 | 0.48±.05 | 0.93±.04 | 0.56±.08 | 1.45±.34 | 0.58±.18 |
| | GMAN | 1.16±.03 | 0.53±.02 | 0.92±.02 | 0.53±.06 | 1.56±.07 | 0.84±.06 | 1.07±.06 | 0.71±.04 | 2.84±1.11 | 0.53±.04 | 3.36±1.36 | 0.47±.09 | 0.98±.04 | 0.52±.05 | 1.30±.07 | 0.55±.06 |
| | MoSSL | 1.10±.07 | 0.45±.07 | 0.90±.11 | 0.54±.14 | 0.98±.03 | 0.63±.31 | 0.75±.02 | 0.58±.26 | 2.62±.12 | 0.65±.07 | 2.43±.06 | 0.55±.03 | 0.74±.02 | 0.41±.01 | 1.02±.04 | 0.40±.01 |
| | DeepCrime | 1.27±.14 | 0.57±.09 | 0.94±.06 | 0.59±.07 | 1.36±.05 | 0.59±.18 | 0.82±.07 | 0.45±.09 | 2.75±.25 | 0.50±.17 | 2.53±.38 | 0.57±.01 | 0.70±.05 | 0.41±.02 | 1.33±.05 | 0.49±.08 |
| | ST-SHN | 1.07±.05 | 0.67±.08 | 0.90±.01 | 0.78±.02 | 1.32±.03 | 0.51±.14 | 0.75±.02 | 0.35±.06 | 2.80±.45 | 0.52±.22 | 2.88±.58 | 0.47±.15 | 0.72±.03 | 0.35±.06 | 1.14±.02 | 0.47±.08 |
| | ST-HSL | 1.13±.01 | 0.50±.02 | 0.94±.01 | 0.52±.03 | 1.25±.12 | 0.45±.09 | 0.77±.10 | 0.44±.06 | 2.62±1.90 | 0.49±.09 | 2.49±.42 | 0.49±.20 | 0.64±.10 | 0.42±.06 | 1.08±.15 | 0.45±.03 |
| *Online* | DLF | 2.89±.17 | 0.57±.03 | 2.04±.11 | 0.55±.02 | 1.48±.08 | 0.40±.03 | 0.88±.07 | 0.33±.14 | 2.93±.78 | 0.42±.36 | 2.65±.06 | 0.49±.13 | 1.05±.04 | 0.33±.14 | 1.94±.06 | 0.49±.03 |
| | FSNet | 2.24±.18 | 0.60±.11 | 1.99±.36 | 0.53±.08 | 1.34±.12 | 0.51±.06 | 0.96±.15 | 0.44±.08 | 2.89±.54 | 0.37±.28 | 2.54±.37 | 0.48±.08 | 0.78±.13 | 0.39±.08 | 1.38±.16 | 0.47±.05 |
| | OneNet | 2.53±.12 | 0.58±.07 | 1.53±.18 | 0.54±.11 | 1.23±.18 | 0.52±.09 | 0.82±.06 | 0.47±.01 | 2.88±.43 | 0.40±.15 | 3.04±.40 | 0.87±.19 | 0.94±.06 | 0.43±.08 | 1.46±.17 | 0.48±.04 |
| | ST-HHOL (Ours) | **0.95±.01** | **0.43±.01** | **0.87±.01** | **0.46±.02** | **0.97±.01** | **0.35±.01** | **0.66±.02** | **0.29±.01** | **2.54±.08** | **0.36±.01** | **2.34±.02** | **0.47±.08** | **0.58±.01** | **0.31±.01** | **0.96±.01** | **0.39±.01** |

## 5.1 COMPARISON TO STATE-OF-THE-ART METHODS (RQ1)

Table 2 and Table 3 summarize ST-HHOL's performance on *crime quantity* and *occurrence prediction*. ST-HHOL consistently outperforms all baselines across all datasets, with complete results and visualizations provided in Appendix E.1 and E.5. For quantity prediction, ST-HHOL reduces average MAE and MAPE by 5.37% and 9.21% on CHI, 3.52% and 8.83% on NYC, 2.97% and 5.85% on PHI, and notably 6.45% and 11.32% on TOR dataset, demonstrating its effectiveness across diverse urban environments. Notably, converting the offline model to an online learning paradigm further improves performance on non-stationary crime data.

In occurrence prediction, ST-HHOL achieves gains of 0.94% (Micro-F1), 1.09% (Macro-F1), and 1.69% (TZR) on NYC; 0.70% (Micro-F1), 0.70% (Macro-F1), and 3.08% (TZR) on CHI. The consistent superiority in True Zero Rate (TZR) across all datasets particularly confirms ST-HHOL's robustness in handling data sparsity and skewed distributions, effectively identifying true zero-occurrence scenarios that are prevalent in crime prediction tasks.

In addition, we evaluate robustness, scalability, complexity, and execution efficiency, with detailed results reported in Appendix E.2–E.4.

Table 3: Comparison of crime occurrence prediction over NYC and CHI datasets.

| Dataset | Metric | SVM | ARIMA | DCRNN | STGCN | DeepCrime | GMAN | ST-SHN | OneNet | DLF | ST-HHOL |
|---------|--------|-----|-------|-------|-------|-----------|------|--------|--------|-----|---------|
| | Micro-F1 ↑ | 0.478 | 0.452 | 0.562 | 0.569 | 0.575 | 0.553 | 0.635 | 0.602 | 0.638 | **0.644** |
| NYC | Macro-F1 ↑ | 0.493 | 0.468 | 0.570 | 0.573 | 0.580 | 0.556 | 0.636 | 0.606 | 0.640 | **0.647** |
| | TZR ↑ | 0.482 | 0.475 | 0.564 | 0.558 | 0.589 | 0.588 | 0.647 | 0.615 | 0.652 | **0.663** |
| | Micro-F1 ↑ | 0.608 | 0.565 | 0.648 | 0.678 | 0.663 | 0.679 | 0.710 | 0.661 | 0.692 | **0.715** |
| CHI | Macro-F1 ↑ | 0.606 | 0.574 | 0.649 | 0.679 | 0.669 | 0.681 | 0.712 | 0.664 | 0.693 | **0.717** |
| | TZR ↑ | 0.603 | 0.579 | 0.637 | 0.675 | 0.676 | 0.669 | 0.714 | 0.673 | 0.688 | **0.736** |

## 5.2 ABLATION STUDY (RQ2)

To comprehensively assess the contribution of each component in ST-HHOL, we compare several variant models as follows: **(1) w/o** $\mathcal{G}^e$: removes the multi-source input and the heterogeneous hypergraph in HHGCN; **(2) w/o** $\mathcal{G}^o$: discards the homogeneous hypergraph in HHGCN; **(3) w/o** $\mathbf{E}_{\mathcal{T}}$: inputs only crime patterns into the spatio-temporal dependency learner, excluding temporal information; **(4) w/o PF-LLM**: removes the PF-LLM component and replaces it with a standard Transformer. **(5) w/o OL**: removes the online learning strategy and reverts to the standard offline setting. The ablation study results over four datasets are shown in Figure 4.

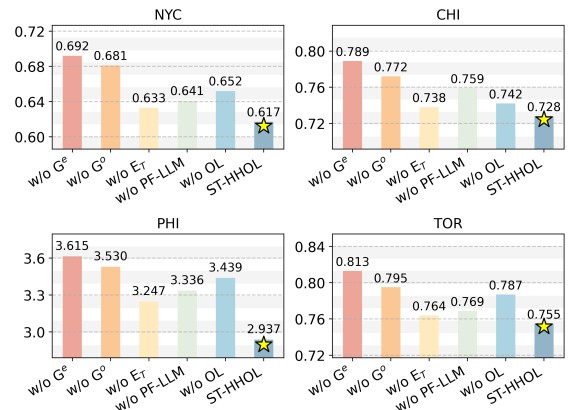

Figure 4: Ablation experiment results.

Each component contributes to the overall performance of ST-HHOL. $\mathcal{G}^e$ and $\mathcal{G}^o$ enhance accuracy by capturing crime-specific and co-occurrence dependencies, respectively. The partially frozen strategy leads to a more efficient transfer of few-shot reasoning ability than a standard Transformer trained from scratch. Moreover, the incorporated online learning mechanism enables the model to adapt to concept drift in evolving crime patterns continuously.

We also compare several PF-LLM variants: **(1)** Frozen Pretrained Transformer (**FPT**); **(2)** models without pretraining (**No Pretrain**); **(3)** fully tuned models with no frozen layers (**Full Tuning**); **(4)** freezing only attention modules (**PF-A**); and **(5)** freezing only FFN layers (**PF-FFN**). As shown in Figure 5, FPT exhibits limited adaptability, whereas Full Tuning reduces errors but incurs higher variance (RMSE), indicating overfitting on sparse data. By contrast, PF-FFN achieves a better trade-off between retaining pretrained knowledge and adapting to the target domain.

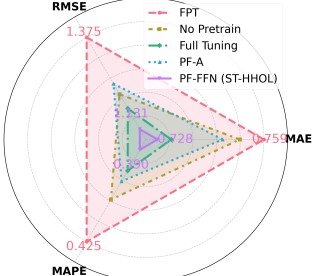

Figure 5: Comparison results for different variants of PF-LLM.

## 5.3 Time Adjustment Analysis (RQ3) & Hyperparameter Study (RQ4)

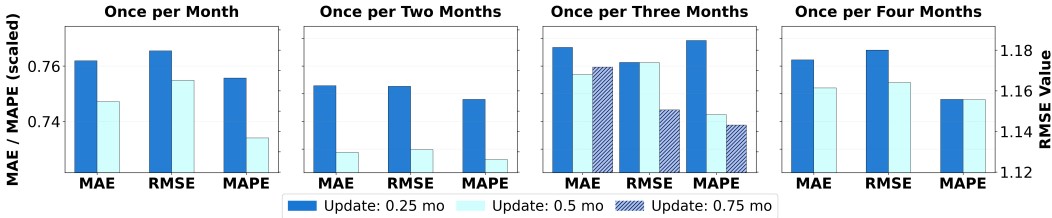

Figure 6: The time adjustment study on the frequency of the fine-tuning and retraining. For intuitive comparison, we scale MAPE to the same vertical axis as MAE.

To assess the impact of retraining and fine-tuning frequency in online learning, we evaluate ST-HHOL under retraining intervals of $\{1, 2, 3, 4\}$ months and fine-tuning intervals of $\{0.25, 0.5, 0.75\}$ months (the latter used only with a 3-month retraining interval). To justify these choices, we first conduct an FFT-based decomposition of the crime series. The analysis reveals pronounced periodic components at two scales: (1) **short-term cycles** within **1–3 weeks**, reflecting fast behavioral fluctuations, and (2) **long-term cycles** spanning **1–4 months**, corresponding to slower structural changes. Such multiscale periodicity aligns with the known nature of concept drift in spatio-temporal crime data. Consequently, the retraining intervals $T \in \{1, 2, 3, 4\}$ months are designed to capture longer-term drift, while the fine-tuning intervals $\tau \in \{0.25, 0.5, 0.75\}$ months (weekly–triweekly) are chosen to align with short-term dynamics.

Experiments on the CHI dataset (Figure 6) report average results across all crime types. Notably, biweekly fine-tuning (0.5 months) consistently outperforms weekly updates (0.25 months), indicating that crime dynamics evolve over a roughly two-week horizon, while overly frequent updates may induce catastrophic forgetting. A two-month retraining interval offers the best trade-off between adaptability and stability. Although 0.75-month fine-tuning slightly improves RMSE and MAPE under the 3-month retraining setting, it does not reduce MAE, limiting its utility. Overall, combining two-month retraining with biweekly fine-tuning proves to be the most effective strategy.

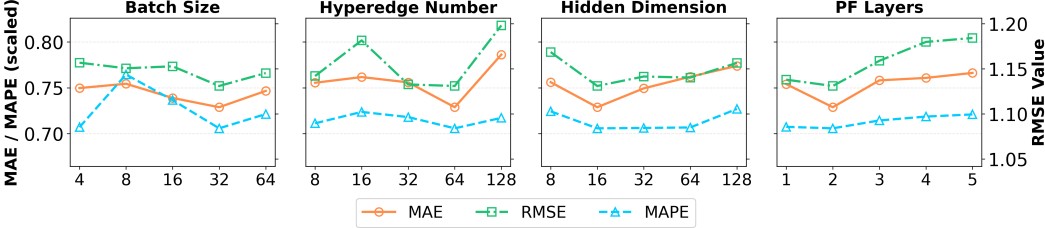

Figure 7: The impact study of different hyperparameter settings over the CHI dataset. Average MAE, RMSE, and MAPE (scaled to align with MAE axis) are reported across all crime types.

We conduct a sensitivity study to evaluate the impact of key hyperparameters of ST-HHOL using the CHI dataset, as shown in Figure 7. Specifically: **(1)** `Batch Size`: Among $\{4, 8, 16, 32, 64\}$, a batch size of 32 yields the lowest prediction error. **(2)** `Hyperedge Number`: Varying the number of hyperedges ($H_o$) in $\mathcal{G}^o$ among $\{8, 16, 32, 64, 128\}$. Setting to 64 yields the best performance, while increasing to 128 introduces redundancy and increases error. **(3)** `Hidden Dimension`: We explore hidden dimensions of features among $\{8, 16, 32, 64, 128\}$. Setting to 16 offers the best performance, while overly high dimensions may amplify prediction errors, such as MAE. **(4)** `PF Layers`: We evaluate the number of partially frozen layers in the PF-LLM from 1 to 5. Setting to 2 offers the optimal trade-off across metrics, while unfreezing more than two layers leads to overfitting under sparse and long-tailed crime data, disrupting pretrained inductive biases.

## 5.4 Case Study (RQ5)

We visualize the crime co-occurrence relationships captured by the homogeneous hypergraph and the heterogeneous influences of multi-source data constructed by the heterogeneous hypergraph in Figures 8 and 9, respectively. Key insights include: **(1)** ST-HHOL captures complex crime co-occurrence patterns. Hyperedge $e_8$ links low-frequency regions, $e_{16}$ and $e_{37}$ cover mid-frequency

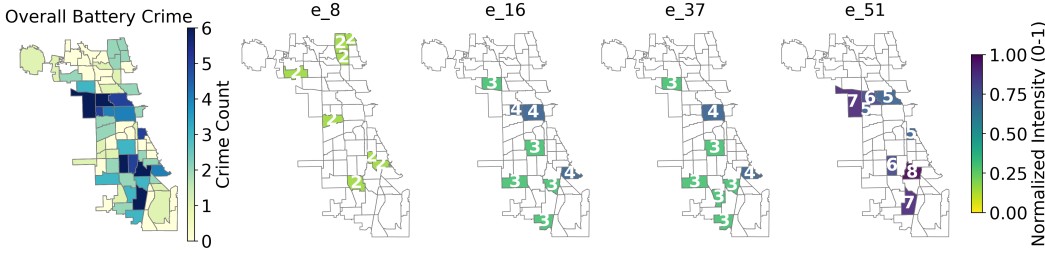

Figure 8: The visualization of the homogeneous hyperedge of crime co-occurrence relations, taking Battery on May 30, 2024 in Chicago as an example.



Figure 9: The visualization of the impact of multi-source factors on crime patterns across types and regions, captured by heterogeneous hyperedges. (CDOT: traffic order, DOB: building violations, Rest: restaurants, Rail: railway stations, Une: unemployment rate, Tem: temperature, Hum: humidity)

areas, and $e_{51}$ clusters high-frequency regions. **(2)** ST-HHOL reflects cross-region and cross-crime type heterogeneity. Crimes in the *Loop* are shaped by restaurant and station densities, while in low-income areas like *Austin*, the unemployment rate is the dominant factor. **(3)** ST-HHOL effectively adapts to the temporal evolution. Compared to December, elevated temperatures in June significantly intensify Theft and Assault in commercial zones such as *Loop* and *Near South*.

## 6 CONCLUSION AND LIMITATIONS

In this paper, we propose **ST-HHOL**, an online spatio-temporal learning framework that models crime dynamics with a hierarchical hypergraph and adapts to non-stationary environments. Leveraging PF-LLM further enhances its ability to learn from sparse and evolving dependencies. Experiments on multiple urban crime datasets show that ST-HHOL achieves superior accuracy, robustness, and interpretability, while offering insights into the dynamics of crime occurrence.

We identify two main directions for future work. First, although ST-HHOL performs well on four diverse urban datasets, its generalizability warrants evaluation across a broader range of cities with heterogeneous urban structures and socioeconomic conditions as more high-quality streaming data become available. Second, this study focuses on structured spatio-temporal features; integrating multimodal signals, such as textual reports, video, or social media streams, could uncover richer latent factors and improve the modeling of complex crime dynamics.

## 7 ETHICS STATEMENT

The goal of ST-HHOL is to analyze long-term and latent spatio-temporal crime patterns in streaming data, rather than to support real-time operational decisions or individual-level risk assessment. The framework operates exclusively on region-level aggregated crime statistics and does not incorporate individuals, demographic attributes, or law enforcement resources. Consequently, its outputs characterize relative temporal and spatial trends across regions and crime types, instead of actionable signals for targeted intervention, which structurally limits risks of biased profiling or over-policing.

We evaluate ST-HHOL across regions with heterogeneous socioeconomic characteristics and observe consistent predictive behavior, indicating that the learned representations are driven primarily by spatio-temporal structure rather than latent demographic bias. Under these constraints, ST-HHOL is intended as an analytical tool for understanding urban crime dynamics, not for prescriptive or punitive decision-making.

ACKNOWLEDGMENT

This research was conducted by the ARC Centre of Excellence for Automated Decision-Making and Society (ADM+S, CE200100005), and funded fully by the Australian Government through the Australian Research Council.

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

# Supplementary Material

## ST-HHOL: Spatio-Temporal Hierarchical Hypergraph Online Learning for Crime Prediction

TABLE OF CONTENTS

## A    NOTATIONS

Key notations used in the paper and their definitions are summarized in Table 4.

Table 4: Main notations and their definitions.

| Notation | Definition |
| --- | --- |
| $\mathbf{X}^t$ | the crime data at time $t$, $\mathbf{X}^t \in \mathbb{R}^{N \times C}$ |
| $\mathbf{S}^t$ | the auxiliary data at time $t$, $\mathbf{S}^t \in \mathbb{R}^{N \times M}$ |
| $N$ | the number of regions |
| $C$ | the number of crime types |
| $M$ | the number of auxiliary data |
| $\mathbf{E}^t$ | the crime patterns at time $t$, $\mathbf{E}^t \in \mathbb{R}^{N \times C}$ |
| $\mathcal{G}^e$ | the heterogeneous hypergraph of HHGCN |
| $\mathcal{G}^o$ | the homogeneous hypergraph of HHGCN |
| $\mathbf{\Theta}_e^t$ | the incident matrix of $\mathcal{G}^e$ at time $t$ |
| $\mathbf{\Theta}_o^t$ | the incident matrix of $\mathcal{G}^o$ at time $t$ |
| $H_e, H_o$ | the number of hyperedges in $\mathcal{G}^e$ and $\mathcal{G}^o$ |

## B    EXTENDED RELATED WORK

**Online Learning.** Online learning has emerged as an effective paradigm for handling concept drift in streaming data (Tsymbal, 2004), enabling models to adapt to evolving data distributions through continual updates. This capability makes it particularly suitable for real-time applications such as recommendation systems (Song et al., 2014; Zhou et al., 2019) and network security detection (Yu et al., 2021). Recent efforts have focused on developing adaptive update strategies that trade off knowledge retention and model plasticity, especially in long-term forecasting tasks. For example, Ddg-da (Li et al., 2022a), FSNet (Pham et al., 2023), and LSTD (Cai et al., 2025) explore different mechanisms to maintain stable performance over time. OneNet (Wen et al., 2023) proposes a

dynamic ensemble strategy that adjusts the weights of dual-stream models based on the distribution of streaming data, rather than relying on a single model.

In the context of urban spatio-temporal data, concept drift often manifests gradually. DOST (Wang et al., 2025) addresses this by employing an awake-hibernate learning strategy to adapt to non-stationary patterns. Similarly, DLF (Wang et al., 2024) decomposes time series into seasonal components and latent trends, using iterative updates to track evolving dynamics. However, crime data present additional challenges due to their inherently abrupt fluctuations—sudden surges or drops that offline models trained solely on historical data struggle to capture. To address these challenges, we propose an online learning framework, ST-HHOL, specifically designed for crime prediction. ST-HHOL iteratively fine-tunes to capture short-term fluctuations, while periodically retraining to adapt to long-term gradual shifts in crime patterns. The detailed description of the ST-HHOL is presented in Section 4.

## C  MORE DETAILS OF HYPERGRAPH

A hypergraph (Sun et al., 2021) can be defined as $\mathcal{G} = \{\mathcal{V}, \mathcal{E}, \mathcal{T}_v, \mathcal{T}_e\}$. Here, $\mathcal{V}$ is the set of vertices and $\mathcal{E}$ is the set of hyperedges. $\mathcal{T}_v$ and $\mathcal{T}_e$ denote the sets of vertex types and hyperedge types, respectively. If $|\mathcal{T}_v| + |\mathcal{T}_e| > 2$, the hypergraph is heterogeneous. Otherwise, the hypergraph is homogeneous. Compared to a pairwise graph, in which each edge connects two vertices, a hyperedge in a hypergraph can connect more than two vertices. For any hyperedge $e \in \mathcal{E}$, it can be denoted as $e = \{v_i, v_j, \ldots, v_k\} \subseteq \mathcal{V}$. A positive diagonal matrix $\mathbf{W} \in \mathbb{R}^{|\mathcal{E}| \times |\mathcal{E}|}$ denotes the hyperedge weights. The relationship between vertices and hyperedges can be represented by an incidence matrix $\mathbf{H} \in \mathbb{R}^{|\mathcal{V}| \times |\mathcal{E}|}$ with entries defined as:

$$H_{v,e} = \begin{cases} 1, & \text{if } v \in e, \\ 0, & \text{otherwise.} \end{cases} \tag{13}$$

Let $\mathbf{D}_v \in \mathbb{R}^{|\mathcal{V}| \times |\mathcal{V}|}$ and $\mathbf{D}_e \in \mathbb{R}^{|\mathcal{E}| \times |\mathcal{E}|}$ denote the diagonal matrices containing the vertex degrees and hyperedge degrees, respectively, where $(\mathbf{D}_v)_{ii} = \sum_{e \in \mathcal{E}} W_e H_{i,e}$, and $(\mathbf{D}_e)_{jj} = \sum_{v \in \mathcal{V}} H_{v,j}$. The hypergraph convolution operator can be defined as $\mathbf{\Theta} = \sigma\left(\mathbf{D}_v^{-\frac{1}{2}} \mathbf{HWD}_e^{-\frac{1}{2}} \mathbf{H}^\top \mathbf{D}_v^{-\frac{1}{2}}\right)$, and the hypergraph Laplacian can be denoted as $\mathbf{\Delta} = \mathbf{I} - \mathbf{\Theta}$.

## D  MORE DETAILS OF DATASETS AND BASELINES

### D.1  DATASETS DETAILS

We conduct experiments on four real-world urban crime datasets from Chicago (CHI)[1], New York City (NYC)[2], Philadelphia (PHI)[3], and Toronto (TOR)[4], as summarized in Table 5. Each dataset includes four major crime types, with corresponding timestamps and spatial locations. The CHI and PHI datasets contain records for 77 boroughs in Chicago and 6 boroughs in Philadelphia, respectively, spanning from January 1, 2023 to December 31, 2024. The NYC dataset covers 123 police districts from January 1, 2022 to December 31, 2023, while the TOR dataset includes 158 neighborhoods with daily crime records from January 1, 2023 to December 31, 2024. In addition, we collect diverse multi-source contextual data, including dynamic 311 service requests, weather conditions, static POI distributions, and socio-economic indicators. Detailed categories and variables are provided in Table 6.

### D.2  BASELINES DETAILS

We select 14 representative baselines spanning traditional models and state-of-the-art deep learning approaches. The baseline descriptions are organized as follows:

---

[1]Chicago data portal is available at: `https://data.cityofchicago.org/`

[2]NYC open data is available at: `https://opendata.cityofnewyork.us/`

[3]Philadelphia open data is available at: `https://opendataphilly.org/`

[4]Toronto Police Service Open Data: `https://data.torontopolice.on.ca/datasets`

Table 5: Summary of the four urban crime datasets.

| Datasets | Region Num | Time Span | Time Interval | Crime Data |
|---|---|---|---|---|
| CHI | 77 | January 1, 2023 – December 31, 2024 | 1 day | Theft, Battery, Assault, Damage |
| NYC | 123 | January 1, 2023 – December 31, 2024 | 1 day | Larceny, Assault, Mischief, Robbery |
| PHI | 6 | January 1, 2023 – December 31, 2024 | 1 day | Theft, Assault, Vehicle, Mischief |
| TOR | 158 | January 1, 2023 – December 31, 2024 | 1 day | Assault, B&E, Robbery, Theft |

Table 6: The description of variables contained in multi-source data.

| Datasets | 311 Service Types | POI | Weather | Others |
|---|---|---|---|---|
| CHI | Streets and Sanitation
CDOT - Department of Transportation
DOB - Buildings | Restaurant&Café
School&University | Temperature
Humidity
Windspeed | Income
Unemployment rate |
| NYC | Illegal Parking
Noise - Residential
Blocked Driveway | Park&Playground
Hospital
Railway station | | |
| PHI | Abandoned Vehicle
Illegal Dumping
Graffiti Removal | Shopping
Bank, Police
Entertainment | | |
| TOR | Property & Environment
Noise
Roads & Traffic | | | |

- SVM (Mattera & Haykin, 1998): Learns maximum-margin hyperplanes in kernel-induced feature spaces for time series regression.

- ARIMA (Box et al., 2015): Combines autoregressive integration with moving average components to handle non-stationary temporal patterns.

- DCRNN (Li et al., 2017): Models spatial diffusion processes via bidirectional random walks, coupled with a sequence-to-sequence architecture for temporal modeling.

- STGCN (Yu et al., 2018): Establishes spatio-temporal correlations through stacked graph convolutions and temporal gated convolutions, eliminating recurrent units.

- AGCRN (Bai et al., 2020): Automatically infers node-wise dependencies with adaptive graph generation while learning personalized patterns through node-specific parameters.

- MTGNN (Wu et al., 2020b): Designs mix-hop graph diffusion layers with dilated temporal convolutions to capture multi-range spatio-temporal dependencies.

- GMAN (Zheng et al., 2020): Integrates spatial attention and temporal attention in stacked transformer blocks for cross-space-time dependency modeling.

- MoSSL (Deng et al., 2024): Constructs multi-granularity self-supervision tasks to enhance representation learning for temporal, spatial, and feature variations.

- DeepCrime (Huang et al., 2018): Unifies crime embedding learning with hierarchical attention mechanisms over spatio-temporal-categorical dimensions.

- ST-HSL (Li et al., 2022b): Addresses label scarcity through hypergraph structure learning and contrastive self-supervision on region representations.

- ST-SHN (Xia et al., 2021): Models crime category dependencies via multi-channel hypergraph routing in a sequential prediction framework.

- DLF (Wang et al., 2024): Decouples trend-seasonal patterns through frequency domain analysis with momentum-updated dual experts.

- FSNet (Pham et al., 2023): Balances fast adaptation and memory retention via parameter-efficient adapter modules and Hopfield network-based associative memory.

- OneNet (Wen et al., 2023): Dynamically fuses temporal and cross-variable dependency models through a meta-learned architecture controller.

## D.3 Evaluation Metrics Details

**Crime quantity prediction.** To assess the performance of crime quantity prediction, we employ the Mean Absolute Error (MAE), Root Mean Square Error (RMSE), and Mean Absolute Percentage Error (MAPE) as the evaluation metrics. MAE measures the average magnitude of errors in predictions, providing an intuitive sense of the overall prediction accuracy. RMSE penalizes larger errors more heavily, thus highlighting the model's sensitivity to significant deviations. MAPE expresses prediction errors as a percentage, providing a scale-independent evaluation that is particularly useful for comparing performance across different regions or periods.

$$\text{MAE}(y, \hat{y}) = \frac{1}{N} \sum_{n=1}^{N} |y_n - \hat{y}_n|, \tag{14}$$

$$\text{RMSE}(y, \hat{y}) = \sqrt{\frac{1}{N} \sum_{n=1}^{N} (y_n - \hat{y}_n)^2}, \tag{15}$$

$$\text{MAPE}(y, \hat{y}) = \frac{1}{N} \sum_{n=1}^{N} |\frac{y_n - \hat{y}_n}{y_n}|, \tag{16}$$

where $y$ represents the actual value, $\hat{y}$ represents the prediction. $N$ denotes the total number of regions.

**Crime occurrence prediction.** For crime-occurrence prediction (binary: 1 = crime occurs, 0 = no crime), we define the confusion counts over all $N$ samples as:

$$TP = \sum_{n=1}^{N} \mathbf{1}\{y_n = 1 \wedge \hat{y}_n = 1\}, \qquad FP = \sum_{n=1}^{N} \mathbf{1}\{y_n = 0 \wedge \hat{y}_n = 1\},$$

$$FN = \sum_{n=1}^{N} \mathbf{1}\{y_n = 1 \wedge \hat{y}_n = 0\}, \qquad TN = \sum_{n=1}^{N} \mathbf{1}\{y_n = 0 \wedge \hat{y}_n = 0\}, \tag{17}$$

where $y_n \in \{0, 1\}$ is the ground-truth label and $\hat{y}_n \in \{0, 1\}$ is the predicted label.

Precision, recall and F1 for the positive class (crime occurrence) are as follows:

$$\text{Precision}_1 = \frac{TP}{TP + FP}, \qquad \text{Recall}_1 = \frac{TP}{TP + FN}, \qquad \text{F1}_1 = \frac{2 \cdot \text{Precision}_1 \cdot \text{Recall}_1}{\text{Precision}_1 + \text{Recall}_1}. \tag{18}$$

Precision, recall, and F1 for the negative class (no crime) can be computed as follows:

$$\text{Precision}_0 = \frac{TN}{TN + FN}, \qquad \text{Recall}_0 = \frac{TN}{TN + FP}, \qquad \text{F1}_0 = \frac{2 \cdot \text{Precision}_0 \cdot \text{Recall}_0}{\text{Precision}_0 + \text{Recall}_0}. \tag{19}$$

In total, Macro-F1 and Micro-F1 can be defined as:

$$\text{Macro-F1} = \frac{\text{F1}_1 + \text{F1}_0}{2},$$

$$\text{Micro-F1} = \frac{2 \sum_{c \in \{0,1\}} TP_c}{2 \sum_c TP_c + \sum_c FP_c + \sum_c FN_c} = \frac{TP + TN}{N}. \tag{20}$$

## D.4 Experimental Settings

All experiments are conducted on a server equipped with an NVIDIA RTX 3090 GPU using PyTorch 2.0. We implement ST-HHOL and all baseline models in a unified framework to ensure fair comparisons. We use the Adam optimizer with an initial learning rate of $1 \times 10^{-3}$, a decay factor of $1 \times 10^{-4}$ applied periodically, and a batch size of 32. Model training is conducted for a maximum of 100 epochs with an early stopping strategy triggered if validation performance does not improve for

10 consecutive epochs. The dataset is chronologically split into a warm-up and an online training phase in a 25:75 ratio. The warm-up phase is further divided into training and validation subsets in a 7:3 ratio. Following prior studies (Li et al., 2022b; Liang et al., 2024; Wu et al., 2024), we set the temporal input length to 7 and the forecasting horizon to 1. Each deep learning model is trained and evaluated across five independent runs, and the final performance is reported as the mean of the evaluation metrics. For ST-HHOL, the loss function includes two balancing coefficients $\lambda_1$ and $\lambda_2$, both set to 0.1. The core component of the spatio-temporal dependency learner PF-LLM, is instantiated with a GPT-2 small architecture[5] with two layers, a hidden size of 768, and 12 attention heads. The internal hidden dimension of ST-HHOL is set to 16, and the number of hyperedges in the homogeneous hypergraph $\mathcal{G}^o$ is set to 64.

# E  MORE EXPERIMENTAL RESULTS

## E.1  COMPLETE COMPARISON RESULTS

For fair comparison, except for the online models DLF, FSNet, and OneNet, we transform all offline baseline models into an online version. Specifically, these models are periodically retrained using new streaming data after the initial warm-up phase. Tables 7, 8, 9, and 10 compare ST-HHOL with a variety of baselines across several crime types and datasets. ST-HHOL consistently outperforms all baselines across different datasets and metrics. For quantity prediction, it reduces average MAE and MAPE by 5.37% and 9.21% on CHI, 3.52% and 8.83% on NYC, 2.97% and 5.85% on PHI, and 4.12% and 7.94% on TOR, demonstrating its effectiveness and generalizability.

Moreover, we summarize the key findings as follows: **(1)** ST-HHOL consistently outperforms all baselines across crime types, metrics, and datasets. This superior performance is attributed to its hierarchical hypergraph design, which effectively models spatial and criminal specificity, thereby enhancing robustness and generalization. **(2)** ST-HHOL achieves significantly lower MAPE than the second-best models, demonstrating a stronger capability to accurately detect non-zero crime occurrences. This indicates that ST-HHOL is more effective at uncovering latent crime patterns and generating precise predictions. **(3)** The online learning strategy tailored in ST-HHOL effectively addresses concept drift, outperforming methods like FSNet, OneNet, and DLF, whose update mechanisms are not optimized for the dynamic and heterogeneous nature of crime data. Moreover, FSNet and OneNet, originally designed for long-term sequences, struggle to adapt to highly varied crime types, limiting their effectiveness in evolving scenarios. **(4)** Existing spatio-temporal models, such as DCRNN and STGCN, can capture temporal dependencies but often underperform when facing crime pattern heterogeneity and skewed distribution. For instance, while they achieve competitive performance on Assault in NYC, they struggle on several high-frequency crimes like Theft and Battery in CHI. In summary, ST-HHOL not only delivers superior predictive performance but also demonstrates enhanced stability across diverse crime categories and spatial distributions.

Table 7: Overall performance of crime prediction on CHI dataset. The results are 5-run error comparison, the **bold** / underlined font means the best / the second-best result.

| Method | Theft | | | Battery | | | Assault | | | Damage | | |
|---|---|---|---|---|---|---|---|---|---|---|---|---|
| | MAE | RMSE | MAPE | MAE | RMSE | MAPE | MAE | RMSE | MAPE | MAE | RMSE | MAPE |
| SVM | 1.54 | 1.75 | 0.51 | 0.94 | 1.89 | 0.56 | 0.60 | 0.98 | 0.51 | 0.69 | 0.94 | 0.52 |
| ARIMA | 1.52 | 1.76 | 0.52 | 0.99 | 1.72 | 0.58 | 0.61 | 0.91 | 0.54 | 0.69 | 0.96 | 0.53 |
| DCRNN | 2.91 ±0.25 | 3.94 ±0.55 | 0.49 ±0.05 | 1.82 ±0.15 | 3.17 ±0.30 | 0.64 ±0.05 | 0.66 ±0.05 | 0.98 ±0.14 | 0.37 ±0.08 | 0.85 ±0.11 | 1.75 ±0.31 | 0.46 ±0.10 |
| STGCN | 2.82 ±0.58 | 3.66 ±0.73 | 0.58 ±0.09 | 2.03 ±0.04 | 2.74 ±0.40 | 0.61 ±0.03 | 0.80 ±0.01 | 1.45 ±0.13 | 0.45 ±0.04 | 1.04 ±0.02 | 2.19 ±0.09 | 0.57 ±0.09 |
| AGCRN | 0.99 ±0.11 | 1.53 ±0.35 | 0.50 ±0.07 | 0.88 ±0.17 | 1.31 ±0.28 | 0.51 ±0.02 | 0.54 ±0.09 | 0.88 ±0.12 | 0.66 ±0.03 | 0.65 ±0.02 | 0.99 ±0.17 | 0.57 ±0.15 |
| MTGNN | 1.45 ±0.34 | 2.34 ±0.56 | 0.58 ±0.18 | 1.06 ±0.33 | 1.56 ±0.17 | 0.60 ±0.08 | 0.86 ±0.06 | 1.04 ±0.07 | 0.70 ±0.08 | 0.90 ±0.11 | 1.11 ±0.20 | 0.71 ±0.14 |
| GMAN | 1.16 ±0.03 | 2.12 ±0.11 | 0.53 ±0.02 | 0.92 ±0.02 | 1.44 ±0.09 | 0.53 ±0.06 | 0.74 ±0.04 | 1.45 ±0.05 | 0.62 ±0.03 | 0.89 ±0.03 | 1.31 ±0.09 | 0.65 ±0.11 |
| MoSSL | 1.10 ±0.07 | 1.67 ±0.01 | 0.45 ±0.07 | 0.90 ±0.11 | 1.33 ±0.13 | 0.54 ±0.14 | 0.64 ±0.01 | 0.85 ±0.01 | 0.49 ±0.18 | 0.70 ±0.02 | 0.92 ±0.05 | 0.43 ±0.11 |
| DeepCrime | 1.27 ±0.14 | 1.66 ±0.28 | 0.57 ±0.09 | 0.94 ±0.06 | 1.34 ±0.27 | 0.59 ±0.07 | 0.68 ±0.09 | 0.94 ±0.15 | 0.54 ±0.11 | 0.69 ±0.22 | 1.04 ±0.18 | 0.65 ±0.10 |
| ST-SHN | 1.07 ±0.05 | 1.63 ±0.27 | 0.67 ±0.08 | 0.90 ±0.01 | 1.35 ±0.21 | 0.78 ±0.02 | 0.69 ±0.03 | 0.90 ±0.01 | 0.52 ±0.04 | 0.68 ±0.16 | 1.03 ±0.10 | 0.67 ±0.05 |
| ST-HSL | 1.13 ±0.01 | 1.68 ±0.01 | 0.50 ±0.02 | 0.94 ±0.01 | 1.31 ±0.01 | 0.52 ±0.03 | 0.67 ±0.01 | 0.93 ±0.02 | 0.42 ±0.00 | 0.72 ±0.03 | 1.03 ±0.07 | 0.47 ±0.02 |
| DLF | 2.89 ±0.17 | 4.18 ±0.28 | 0.57 ±0.03 | 2.04 ±0.11 | 2.94 ±0.19 | 0.55 ±0.02 | 1.23 ±0.09 | 1.87 ±0.15 | 0.32 ±0.01 | 1.46 ±0.08 | 2.10 ±0.14 | 0.61 ±0.01 |
| FSNet | 2.24 ±0.18 | 3.17 ±0.58 | 0.60 ±0.11 | 1.99 ±0.36 | 2.38 ±0.15 | 0.53 ±0.08 | 0.99 ±0.12 | 1.12 ±0.18 | 0.45 ±0.05 | 0.96 ±0.08 | 1.20 ±0.10 | 0.63 ±0.03 |
| OneNet | 2.53 ±0.12 | 3.25 ±0.30 | 0.58 ±0.07 | 1.53 ±0.18 | 2.04 ±0.22 | 0.54 ±0.11 | 1.06 ±0.13 | 1.24 ±0.11 | 0.47 ±0.03 | 1.00 ±0.12 | 1.48 ±0.16 | 0.64 ±0.12 |
| ST-HHOL (Ours) | **0.95** ±0.01 | **1.52** ±0.01 | **0.43** ±0.01 | **0.87** ±0.01 | **1.26** ±0.02 | **0.46** ±0.02 | **0.51** ±0.01 | **0.84** ±0.01 | **0.27** ±0.01 | **0.58** ±0.01 | **0.90** ±0.02 | **0.40** ±0.02 |

---

[5]The source code of GPT-2 is available at: `https://huggingface.co/openai-community/gpt2`

Table 8: Overall performance of crime prediction on NYC dataset. The results are 5-run error comparison, the **bold** / underlined font means the best / the second-best result.

| Model | Larceny | | | Assault | | | Mischief | | | Robbery | | |
|---|---|---|---|---|---|---|---|---|---|---|---|---|
| | MAE | RMSE | MAPE | MAE | RMSE | MAPE | MAE | RMSE | MAPE | MAE | RMSE | MAPE |
| SVM | 1.81 | 2.55 | 0.61 | 0.79 | 1.29 | 0.53 | 0.50 | 0.97 | 0.68 | 0.80 | 1.13 | 0.69 |
| ARIMA | 1.17 | 2.58 | 0.63 | 0.75 | 1.21 | 0.50 | 0.52 | 0.76 | 0.61 | 0.89 | 1.16 | 0.63 |
| DCRNN | $1.42_{\pm0.07}$ | $2.67_{\pm0.21}$ | $0.44_{\pm0.03}$ | $0.85_{\pm0.01}$ | $1.60_{\pm0.06}$ | $0.50_{\pm0.04}$ | $\mathbf{0.33}_{\pm0.01}$ | $0.76_{\pm0.16}$ | $0.20_{\pm0.01}$ | $0.62_{\pm0.16}$ | $1.06_{\pm0.25}$ | $0.20_{\pm0.02}$ |
| STGCN | $1.41_{\pm0.03}$ | $2.44_{\pm0.17}$ | $\underline{0.39}_{\pm0.04}$ | $0.86_{\pm0.01}$ | $1.76_{\pm0.18}$ | $0.46_{\pm0.07}$ | $0.35_{\pm0.02}$ | $0.86_{\pm0.22}$ | $\underline{0.17}_{\pm0.01}$ | $0.70_{\pm0.21}$ | $1.14_{\pm0.31}$ | $0.20_{\pm0.01}$ |
| AGCRN | $1.26_{\pm0.39}$ | $2.19_{\pm0.39}$ | $0.74_{\pm0.01}$ | $\underline{0.70}_{\pm0.01}$ | $1.06_{\pm0.05}$ | $\underline{0.30}_{\pm0.26}$ | $0.38_{\pm0.01}$ | $0.83_{\pm0.01}$ | $0.28_{\pm0.22}$ | $\underline{0.52}_{\pm0.09}$ | $1.16_{\pm0.15}$ | $0.31_{\pm0.20}$ |
| MTGNN | $1.44_{\pm0.22}$ | $2.26_{\pm0.41}$ | $0.57_{\pm0.05}$ | $1.09_{\pm0.12}$ | $1.54_{\pm0.16}$ | $0.60_{\pm0.14}$ | $0.85_{\pm0.09}$ | $1.06_{\pm0.08}$ | $0.72_{\pm0.05}$ | $0.86_{\pm0.12}$ | $1.08_{\pm0.16}$ | $0.70_{\pm0.07}$ |
| GMAN | $1.56_{\pm0.07}$ | $2.64_{\pm0.10}$ | $0.84_{\pm0.06}$ | $1.07_{\pm0.06}$ | $1.63_{\pm0.02}$ | $0.71_{\pm0.01}$ | $1.04_{\pm0.09}$ | $1.87_{\pm0.07}$ | $0.75_{\pm0.01}$ | $1.31_{\pm0.07}$ | $1.77_{\pm0.02}$ | $0.87_{\pm0.05}$ |
| MoSSL | $\underline{0.98}_{\pm0.03}$ | $\underline{1.85}_{\pm0.17}$ | $0.63_{\pm0.31}$ | $0.75_{\pm0.02}$ | $\underline{1.06}_{\pm0.01}$ | $0.58_{\pm0.26}$ | $0.56_{\pm0.01}$ | $0.81_{\pm0.04}$ | $0.18_{\pm0.02}$ | $0.54_{\pm0.01}$ | $\underline{1.06}_{\pm0.15}$ | $0.21_{\pm0.03}$ |
| DeepCrime | $1.36_{\pm0.05}$ | $2.64_{\pm0.12}$ | $0.59_{\pm0.18}$ | $0.82_{\pm0.07}$ | $1.18_{\pm0.08}$ | $0.45_{\pm0.09}$ | $0.64_{\pm0.11}$ | $0.88_{\pm0.14}$ | $0.29_{\pm0.09}$ | $0.68_{\pm0.15}$ | $1.30_{\pm0.09}$ | $0.40_{\pm0.07}$ |
| ST-SHN | $1.32_{\pm0.03}$ | $2.16_{\pm0.06}$ | $0.51_{\pm0.14}$ | $0.75_{\pm0.02}$ | $1.10_{\pm0.02}$ | $0.35_{\pm0.06}$ | $0.49_{\pm0.07}$ | $0.84_{\pm0.02}$ | $0.27_{\pm0.07}$ | $0.57_{\pm0.12}$ | $1.12_{\pm0.05}$ | $0.36_{\pm0.05}$ |
| ST-HSL | $1.25_{\pm0.12}$ | $2.35_{\pm0.01}$ | $0.45_{\pm0.09}$ | $0.77_{\pm0.10}$ | $1.21_{\pm0.07}$ | $0.44_{\pm0.06}$ | $0.65_{\pm0.12}$ | $0.95_{\pm0.12}$ | $0.44_{\pm0.07}$ | $0.79_{\pm0.07}$ | $1.18_{\pm0.05}$ | $0.47_{\pm0.03}$ |
| DLF | $1.48_{\pm0.08}$ | $3.25_{\pm0.13}$ | $0.40_{\pm0.03}$ | $0.88_{\pm0.07}$ | $1.94_{\pm0.06}$ | $0.33_{\pm0.14}$ | $0.66_{\pm0.09}$ | $1.44_{\pm0.16}$ | $0.40_{\pm0.21}$ | $1.02_{\pm0.15}$ | $2.01_{\pm0.24}$ | $0.30_{\pm0.20}$ |
| FSNet | $1.34_{\pm0.12}$ | $3.02_{\pm0.17}$ | $0.51_{\pm0.06}$ | $0.96_{\pm0.15}$ | $1.91_{\pm0.06}$ | $0.44_{\pm0.06}$ | $0.87_{\pm0.14}$ | $1.86_{\pm0.19}$ | $0.46_{\pm0.07}$ | $1.12_{\pm0.08}$ | $1.60_{\pm0.09}$ | $0.32_{\pm0.07}$ |
| OneNet | $1.23_{\pm0.18}$ | $2.99_{\pm0.23}$ | $0.52_{\pm0.09}$ | $0.82_{\pm0.06}$ | $1.91_{\pm0.17}$ | $0.47_{\pm0.06}$ | $0.73_{\pm0.08}$ | $1.50_{\pm0.10}$ | $0.46_{\pm0.05}$ | $1.09_{\pm0.07}$ | $1.24_{\pm0.10}$ | $0.35_{\pm0.08}$ |
| ST-HHOL (Ours) | $\mathbf{0.97}_{\pm0.01}$ | $\mathbf{1.84}_{\pm0.01}$ | $\mathbf{0.35}_{\pm0.01}$ | $\mathbf{0.66}_{\pm0.02}$ | $\mathbf{1.05}_{\pm0.10}$ | $\mathbf{0.29}_{\pm0.01}$ | $\underline{0.34}_{\pm0.02}$ | $\mathbf{0.73}_{\pm0.04}$ | $\mathbf{0.15}_{\pm0.01}$ | $\mathbf{0.50}_{\pm0.04}$ | $\mathbf{1.04}_{\pm0.17}$ | $\mathbf{0.18}_{\pm0.01}$ |

Table 9: Overall performance of crime prediction on PHI dataset. The results are 5-run error comparison, the **bold** / underlined font means the best / the second-best result.

| Model | Theft | | | Assault | | | Vehicle | | | Mischief | | |
|---|---|---|---|---|---|---|---|---|---|---|---|---|
| | MAE | RMSE | MAPE | MAE | RMSE | MAPE | MAE | RMSE | MAPE | MAE | RMSE | MAPE |
| SVM | 5.91 | 6.82 | 0.38 | 3.61 | 4.76 | 0.50 | 2.85 | 3.49 | 0.49 | 2.95 | 3.65 | 0.48 |
| ARIMA | 5.68 | 6.89 | 0.41 | 3.89 | 4.87 | 0.42 | 2.62 | 3.78 | 0.49 | 3.34 | 3.72 | 0.58 |
| DCRNN | $5.86_{\pm2.12}$ | $7.52_{\pm3.16}$ | $0.57_{\pm0.01}$ | $3.74_{\pm1.41}$ | $4.61_{\pm2.74}$ | $0.43_{\pm0.01}$ | $3.09_{\pm0.49}$ | $3.59_{\pm2.49}$ | $0.47_{\pm0.05}$ | $3.43_{\pm0.71}$ | $4.52_{\pm2.40}$ | $0.58_{\pm0.01}$ |
| STGCN | $5.45_{\pm2.58}$ | $7.47_{\pm0.32}$ | $0.55_{\pm0.10}$ | $3.60_{\pm2.36}$ | $4.17_{\pm3.68}$ | $0.42_{\pm0.12}$ | $3.54_{\pm1.60}$ | $3.50_{\pm2.32}$ | $0.38_{\pm0.06}$ | $2.78_{\pm1.15}$ | $3.62_{\pm2.04}$ | $0.48_{\pm0.05}$ |
| AGCRN | $5.13_{\pm0.59}$ | $6.62_{\pm0.72}$ | $0.39_{\pm0.03}$ | $3.18_{\pm0.23}$ | $\mathbf{4.03}_{\pm0.21}$ | $0.39_{\pm0.10}$ | $2.68_{\pm0.03}$ | $3.46_{\pm0.12}$ | $0.41_{\pm0.01}$ | $2.70_{\pm0.22}$ | $3.53_{\pm0.27}$ | $0.49_{\pm0.02}$ |
| MTGNN | $5.08_{\pm1.17}$ | $6.11_{\pm1.91}$ | $0.49_{\pm0.08}$ | $3.48_{\pm1.34}$ | $4.37_{\pm2.77}$ | $0.45_{\pm0.12}$ | $\underline{2.59}_{\pm0.83}$ | $4.53_{\pm0.94}$ | $0.48_{\pm0.12}$ | $2.93_{\pm0.98}$ | $3.67_{\pm1.17}$ | $0.48_{\pm0.05}$ |
| GMAN | $5.13_{\pm2.23}$ | $6.15_{\pm2.80}$ | $0.53_{\pm0.13}$ | $3.72_{\pm0.24}$ | $4.05_{\pm0.57}$ | $0.42_{\pm0.04}$ | $\underline{2.84}_{\pm1.11}$ | $3.69_{\pm1.24}$ | $0.53_{\pm0.04}$ | $3.36_{\pm1.36}$ | $4.52_{\pm1.46}$ | $\underline{0.47}_{\pm0.09}$ |
| MoSSL | $4.13_{\pm0.15}$ | $5.27_{\pm0.33}$ | $0.28_{\pm0.01}$ | $\mathbf{3.02}_{\pm0.05}$ | $4.71_{\pm0.39}$ | $\underline{0.35}_{\pm0.01}$ | $2.62_{\pm0.12}$ | $3.51_{\pm0.05}$ | $0.65_{\pm0.07}$ | $2.43_{\pm0.06}$ | $3.36_{\pm0.09}$ | $0.55_{\pm0.03}$ |
| DeepCrime | $4.01_{\pm0.30}$ | $\underline{5.14}_{\pm0.53}$ | $0.31_{\pm0.03}$ | $3.24_{\pm0.08}$ | $4.24_{\pm0.31}$ | $0.40_{\pm0.02}$ | $2.75_{\pm0.25}$ | $3.52_{\pm0.56}$ | $0.50_{\pm0.17}$ | $2.53_{\pm0.38}$ | $\mathbf{3.24}_{\pm0.49}$ | $0.57_{\pm0.01}$ |
| ST-SHN | $5.62_{\pm0.77}$ | $7.34_{\pm0.93}$ | $0.33_{\pm0.05}$ | $3.79_{\pm0.26}$ | $4.86_{\pm0.34}$ | $0.41_{\pm0.08}$ | $2.80_{\pm0.45}$ | $3.57_{\pm0.14}$ | $0.52_{\pm0.22}$ | $2.88_{\pm0.58}$ | $3.73_{\pm0.62}$ | $0.47_{\pm0.15}$ |
| ST-HSL | $\underline{3.96}_{\pm0.97}$ | $5.31_{\pm1.33}$ | $\underline{0.25}_{\pm0.08}$ | $3.16_{\pm1.89}$ | $4.25_{\pm2.13}$ | $0.38_{\pm0.16}$ | $2.62_{\pm1.90}$ | $3.65_{\pm1.97}$ | $0.49_{\pm0.09}$ | $2.49_{\pm0.42}$ | $\underline{3.26}_{\pm0.38}$ | $0.49_{\pm0.20}$ |
| DLF | $6.60_{\pm1.53}$ | $7.78_{\pm2.24}$ | $0.46_{\pm0.12}$ | $3.88_{\pm0.63}$ | $4.92_{\pm1.06}$ | $0.42_{\pm0.11}$ | $2.93_{\pm0.78}$ | $\underline{3.36}_{\pm1.15}$ | $0.42_{\pm0.36}$ | $2.65_{\pm0.66}$ | $3.98_{\pm0.89}$ | $0.49_{\pm0.13}$ |
| FSNet | $5.30_{\pm0.89}$ | $6.19_{\pm1.25}$ | $0.40_{\pm0.09}$ | $3.80_{\pm0.33}$ | $4.64_{\pm0.84}$ | $0.40_{\pm0.09}$ | $2.89_{\pm0.54}$ | $3.43_{\pm0.79}$ | $\underline{0.37}_{\pm0.28}$ | $2.54_{\pm0.37}$ | $3.97_{\pm0.55}$ | $0.48_{\pm0.08}$ |
| OneNet | $4.88_{\pm0.76}$ | $5.47_{\pm1.23}$ | $0.48_{\pm0.07}$ | $3.54_{\pm0.40}$ | $4.57_{\pm0.58}$ | $0.45_{\pm0.08}$ | $2.88_{\pm0.43}$ | $3.42_{\pm0.82}$ | $0.40_{\pm0.15}$ | $3.04_{\pm0.40}$ | $4.15_{\pm0.78}$ | $0.87_{\pm0.19}$ |
| ST-HHOL (Ours) | $\mathbf{3.83}_{\pm0.09}$ | $\mathbf{4.97}_{\pm0.01}$ | $\mathbf{0.22}_{\pm0.01}$ | $\underline{3.04}_{\pm0.05}$ | $\underline{4.05}_{\pm0.18}$ | $\mathbf{0.34}_{\pm0.01}$ | $\mathbf{2.54}_{\pm0.08}$ | $\mathbf{3.35}_{\pm0.02}$ | $\mathbf{0.36}_{\pm0.09}$ | $\mathbf{2.34}_{\pm0.02}$ | $3.22_{\pm0.16}$ | $\mathbf{0.47}_{\pm0.08}$ |

Table 10: Overall performance of crime prediction on TOR dataset. The results are 5-run error comparison, the **bold** / underlined font means the best / the second-best result.

| Method | Assault | | | B&E | | | Robbery | | | Theft | | |
|---|---|---|---|---|---|---|---|---|---|---|---|---|
| | MAE | RMSE | MAPE | MAE | RMSE | MAPE | MAE | RMSE | MAPE | MAE | RMSE | MAPE |
| SVM | 0.74 | 1.17 | 0.53 | 1.62 | 2.04 | 0.56 | 0.83 | 1.15 | 0.67 | 1.54 | 1.77 | 0.54 |
| ARIMA | 0.71 | 1.18 | 0.50 | 1.56 | 2.02 | 0.55 | 0.89 | 1.16 | 0.68 | 1.52 | 1.73 | 0.53 |
| DCRNN | $0.74_{\pm0.05}$ | $1.12_{\pm0.10}$ | $0.44_{\pm0.03}$ | $1.50_{\pm0.10}$ | $2.50_{\pm0.20}$ | $0.45_{\pm0.05}$ | $0.63_{\pm0.07}$ | $1.14_{\pm0.18}$ | $\underline{0.20}_{\pm0.03}$ | $2.13_{\pm0.16}$ | $3.07_{\pm0.24}$ | $0.47_{\pm0.05}$ |
| STGCN | $0.83_{\pm0.02}$ | $1.26_{\pm0.05}$ | $0.49_{\pm0.04}$ | $1.43_{\pm0.06}$ | $2.41_{\pm0.16}$ | $0.43_{\pm0.06}$ | $0.71_{\pm0.22}$ | $1.15_{\pm0.27}$ | $0.21_{\pm0.01}$ | $2.04_{\pm0.44}$ | $2.96_{\pm0.52}$ | $0.50_{\pm0.05}$ |
| AGCRN | $\underline{0.62}_{\pm0.01}$ | $0.97_{\pm0.06}$ | $\underline{0.35}_{\pm0.01}$ | $\underline{0.98}_{\pm0.10}$ | $1.72_{\pm0.35}$ | $0.43_{\pm0.05}$ | $\underline{0.52}_{\pm0.08}$ | $1.12_{\pm0.09}$ | $0.30_{\pm0.07}$ | $\underline{1.01}_{\pm0.09}$ | $\underline{1.70}_{\pm0.35}$ | $0.50_{\pm0.07}$ |
| MTGNN | $0.93_{\pm0.04}$ | $1.32_{\pm0.07}$ | $0.56_{\pm0.08}$ | $1.45_{\pm0.34}$ | $2.30_{\pm0.56}$ | $0.58_{\pm0.18}$ | $0.86_{\pm0.12}$ | $1.28_{\pm0.79}$ | $0.48_{\pm0.05}$ | $1.45_{\pm0.34}$ | $2.34_{\pm0.42}$ | $0.51_{\pm0.17}$ |
| GMAN | $0.98_{\pm0.04}$ | $1.37_{\pm0.05}$ | $0.52_{\pm0.05}$ | $1.30_{\pm0.07}$ | $2.20_{\pm0.10}$ | $0.55_{\pm0.06}$ | $1.31_{\pm0.01}$ | $1.77_{\pm0.05}$ | $0.49_{\pm0.03}$ | $1.30_{\pm0.03}$ | $2.21_{\pm0.12}$ | $0.59_{\pm0.03}$ |
| MoSSL | $0.74_{\pm0.02}$ | $\underline{0.93}_{\pm0.01}$ | $0.41_{\pm0.01}$ | $1.02_{\pm0.04}$ | $\underline{1.71}_{\pm0.02}$ | $\underline{0.40}_{\pm0.01}$ | $0.54_{\pm0.01}$ | $\underline{1.06}_{\pm0.15}$ | $0.21_{\pm0.03}$ | $1.08_{\pm0.05}$ | $1.70_{\pm0.01}$ | $\underline{0.45}_{\pm0.07}$ |
| DeepCrime | $0.70_{\pm0.08}$ | $1.16_{\pm0.06}$ | $0.41_{\pm0.02}$ | $1.33_{\pm0.05}$ | $2.09_{\pm0.15}$ | $0.49_{\pm0.08}$ | $0.68_{\pm0.05}$ | $1.32_{\pm0.49}$ | $0.56_{\pm0.10}$ | $1.27_{\pm0.18}$ | $2.14_{\pm0.35}$ | $0.56_{\pm0.10}$ |
| ST-SHN | $0.72_{\pm0.03}$ | $1.13_{\pm0.03}$ | $0.35_{\pm0.06}$ | $1.14_{\pm0.02}$ | $1.82_{\pm0.05}$ | $0.47_{\pm0.08}$ | $0.57_{\pm0.12}$ | $1.12_{\pm0.05}$ | $0.36_{\pm0.05}$ | $1.21_{\pm0.05}$ | $1.90_{\pm0.27}$ | $0.56_{\pm0.08}$ |
| ST-HSL | $0.64_{\pm0.10}$ | $1.02_{\pm0.05}$ | $0.42_{\pm0.06}$ | $1.08_{\pm0.15}$ | $2.93_{\pm0.01}$ | $0.45_{\pm0.03}$ | $0.79_{\pm0.05}$ | $1.18_{\pm0.05}$ | $0.47_{\pm0.03}$ | $1.14_{\pm0.01}$ | $1.99_{\pm0.01}$ | $0.54_{\pm0.02}$ |
| DLF | $1.05_{\pm0.04}$ | $1.76_{\pm0.05}$ | $0.33_{\pm0.14}$ | $1.94_{\pm0.06}$ | $3.01_{\pm0.10}$ | $0.49_{\pm0.09}$ | $1.02_{\pm0.15}$ | $2.03_{\pm0.20}$ | $0.30_{\pm0.20}$ | $1.76_{\pm0.14}$ | $3.11_{\pm0.14}$ | $0.50_{\pm0.03}$ |
| FSNet | $0.78_{\pm0.13}$ | $1.63_{\pm0.20}$ | $0.39_{\pm0.08}$ | $1.38_{\pm0.16}$ | $2.73_{\pm0.24}$ | $0.47_{\pm0.03}$ | $1.12_{\pm0.08}$ | $1.60_{\pm0.09}$ | $0.32_{\pm0.07}$ | $1.46_{\pm0.13}$ | $3.04_{\pm0.36}$ | $0.54_{\pm0.11}$ |
| OneNet | $0.94_{\pm0.06}$ | $1.91_{\pm0.14}$ | $0.43_{\pm0.08}$ | $1.46_{\pm0.17}$ | $2.92_{\pm0.34}$ | $0.48_{\pm0.04}$ | $1.09_{\pm0.07}$ | $1.24_{\pm0.10}$ | $0.35_{\pm0.08}$ | $1.52_{\pm0.16}$ | $2.75_{\pm0.41}$ | $0.52_{\pm0.04}$ |
| ST-HHOL (Ours) | $\mathbf{0.58}_{\pm0.01}$ | $\mathbf{0.89}_{\pm0.01}$ | $\mathbf{0.31}_{\pm0.01}$ | $\mathbf{0.96}_{\pm0.01}$ | $\mathbf{1.68}_{\pm0.01}$ | $\mathbf{0.39}_{\pm0.01}$ | $\mathbf{0.50}_{\pm0.04}$ | $\mathbf{1.02}_{\pm0.02}$ | $\mathbf{0.18}_{\pm0.01}$ | $\mathbf{0.98}_{\pm0.01}$ | $\mathbf{1.70}_{\pm0.01}$ | $\mathbf{0.40}_{\pm0.01}$ |

## E.2 ROBUSTNESS ANALYSIS

To further assess the robustness of ST-HHOL under data sparsity, we analyze its performance in regions with low crime frequencies. Based on the frequency distribution of different crime types across areas in the test set, we categorize regions into two low-frequency intervals: [0,0.25) and [0.25,0.5). As illustrated in Figure 10, we visualize the MAE of ST-HHOL and several strong baseline models within these intervals, using the CHI dataset as a case study.

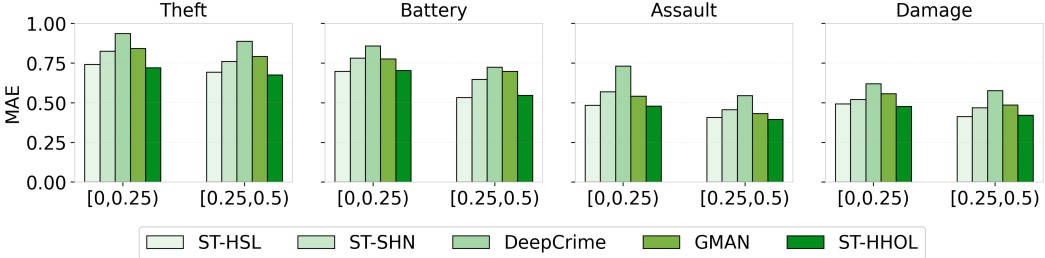

Figure 10: The robustness study of low-frequency crime areas for different crime types.

The results reveal that ST-HHOL maintains superior predictive performance even in sparsely populated urban zones. Despite the inherent challenges posed by imbalanced spatial distributions, ST-HHOL mitigates these limitations by incorporating ubiquitous multi-source data to enrich the sparse crime records. Instead of relying solely on observed data, it uncovers latent crime patterns with spatial and criminal specificity that serve as more informative and generalized crime representations, enabling more reliable predictions in low-frequency regions.

## E.3 SCALABILITY ANALYSIS

To assess the scalability of ST-HHOL, we further evaluate it on the NYC Taxi[6] dataset against several representative spatio-temporal forecasting methods. As shown in Table 11, ST-HHOL achieves competitive results across different variable types and evaluation metrics. Although originally designed for online crime prediction, the underlying motivation of ST-HHOL lies in modeling multivariate spatio-temporal streams with spatial and variable specificity as well as concept drift, which makes it broadly applicable beyond the crime domain.

Table 11: Performance comparison over the NYC Taxi dataset for both pick-up and drop-off prediction. The best and second-best results are highlighted in **bold** and underline, respectively.

| Method | Pick-up | | | Drop-off | | |
|---|---|---|---|---|---|---|
| | MAE | RMSE | MAPE | MAE | RMSE | MAPE |
| DCRNN | 5.40 | 9.71 | 0.35 | 5.19 | 9.63 | 0.37 |
| STGCN | 5.71 | 10.22 | 0.36 | 5.38 | 9.60 | 0.39 |
| AGCRN | 5.79 | 10.11 | 0.40 | 5.45 | 9.56 | 0.40 |
| GMAN | 5.43 | **9.47** | **0.34** | 5.09 | 8.95 | **0.35** |
| ASTGNN | 5.90 | 10.71 | 0.40 | 6.28 | 12.00 | 0.49 |
| ST-HHOL (Ours) | **5.38** | 9.56 | 0.35 | **5.06** | **9.54** | 0.36 |

## E.4 EFFICIENCY ANALYSIS

To compare the computational cost between ST-HHOL and state-of-the-art crime prediction methods, we present their model complexity and execution efficiency, as demonstrated in Table 12. It can be observed that ST-HHOL achieves highly competitive training and inference speeds. Although ST-HHOL comprises 47.39M parameters, approximately 98% of them originate from the spatio-temporal dependency learner (PF-LLM) based on `GPT-2`. The additional components, such as the hierarchical hypergraph structure, impose minimal computational overhead. Moreover, since the parameters of certain modules—such as the feedforward layers of `GPT-2` and the crime-pattern hypergraph fine-tuned during training—are frozen, the actual computational resource consumption of ST-HHOL remains relatively modest.

---

[6]https://www.nyc.gov/site/tlc/about/tlc-trip-record-data.page

Table 12: Complexity and execution efficiency analysis of models over CHI dataset.

| Method | ST-HSL | ST-SHN | DCRNN | STGCN | GMAN | MoSSL | DLF | ST-HHOL (ours) | $-\mathcal{G}^e$ | $-\mathcal{G}^o$ | $-PF-LLM$ |
|---|---|---|---|---|---|---|---|---|---|---|---|
| # Parameters (M) | 0.378 | 0.012 | 0.377 | 0.422 | 0.210 | 1.036 | 0.092 | 47.391 | 0.23 | 0.14 | 46.50 |
| GPU Memory (MB) | 114.98 | 76.98 | 290.78 | 253.57 | 185.95 | 272.08 | 162.53 | 248.52 | 24.13 | 13.25 | 198.80 |
| Training cost (epoch) | 0.14s | 99.62s | 1.06s | 0.66s | 0.38s | 0.47s | 0.27s | 0.44s | 0.045 | 0.035 | 0.360 |
| Test cost (epoch) | 0.03s | 14.47s | 0.14s | 0.14s | 0.06s | 0.11s | 0.04s | 0.09s | 0.007 | 0.005 | 0.078 |
| Average MAE | 0.87 | 0.84 | 1.58 | 1.67 | 0.95 | 0.83 | 1.91 | **0.73** | - | - | - |
| Average RMSE | 1.23 | 1.22 | 2.47 | 2.53 | 1.54 | 1.18 | 2.77 | **1.13** | - | - | - |

### E.5 VISUALIZATION OF PREDICTION RESULTS

To intuitively illustrate the discrepancy between ST-HHOL's predictions and the ground truth, we visualize several representative examples in Figure 11. These heatmaps span multiple time periods and crime categories. As observed, the high consistency between the prediction results and actual spatial distributions of different crime occurrences underscores ST-HHOL's outstanding predictive performance.

Overall, ST-HHOL effectively models both regional differences and category-specific variations in crime intensity. It accurately distinguishes high-frequency crime regions and estimates threshold effects with notable granularity. Moreover, its online learning mechanism continually updates model parameters in response to streaming data, allowing the model to remain responsive to short-term fluctuations and long-term gradual shifts in crime distributions. As a result, ST-HHOL maintains stable forecasting quality over extended time horizons. These results collectively demonstrate the model's robustness and practical utility in real-world, temporally evolving environments.

## F THE USE OF LARGE LANGUAGE MODELS (LLMS)

LLMs were used solely as a general-purpose assistive tool for language polishing and improving the clarity and readability of the manuscript. The authors independently developed all scientific ideas, experiments, analyses, and results. The LLM did not contribute to any research ideation, methodology design, or interpretation of results.

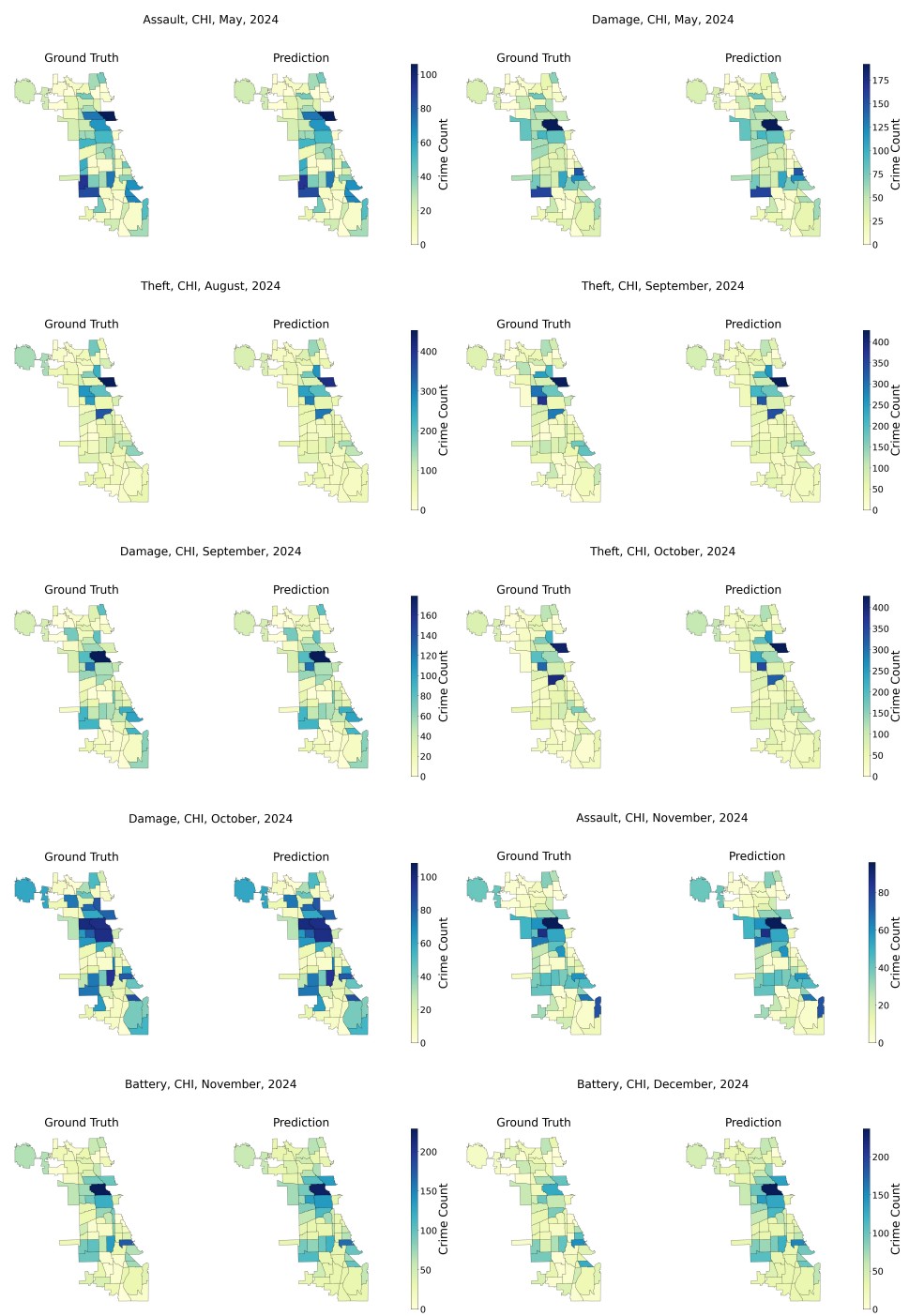

Figure 11: The visualization of prediction results and ground truth.

