# OpenReview forum: "ST-HHOL: Spatio-Temporal Hierarchical Hypergraph Online Learning for Crime Prediction"
_ICLR.cc/2026/Conference — ICLR 2026 Poster_

### Official Review · Reviewer_3QXX · 2025-10-31

**Soundness:** 2
**Presentation:** 3
**Contribution:** 2
**Rating:** 4
**Confidence:** 4

**Summary:**

In this paper, the authors designed a online learning model for crime prediction, and it has a good performance in three datasets.

**Strengths:**

1. A very clear introduction about the methodology.
2. It is innovative to include online learning in this problem.
3. Numerical results show that this model is competitive.
4. The case study is illustrative and useful.

**Weaknesses:**

1. Generally, there are many zeros in crime data; how do this model address this problem?
2. True zero rate is an important metric; however, it is overlooked in this paper.
3. Importance of e multi-source auxiliary data is not comprehensively discussed. Additionally, ablation study of the other two datasets should also be included since the importance of online learning should be shown.

**Questions:**

See weaknesses.

---

> ### Author Response · Authors · 2025-11-17
> **Response to Reviewer 3QXX**
>
> Dear Review 3QXX,
>
> We sincerely thank you for your hard work on our paper. We carefully considered each of your comments and dealt with them accordingly.
>
>
>
> > **Q1: Generally, there are many zeros in crime data; how do this model address this problem?**
> >
>
> A1: ST-HHOL addresses data sparsity through three key components:
>
> **(1) Heterogeneous multi-source contextual integration.**
> Motivated by the inductive biases in crime generation, ST-HHOL anchors each region–crime pair to dense external drivers (POI, 311, weather, socioeconomic data) and integrates them with heterogeneous weights. This preserves crime-type–specific semantics and enriches sparse observations, avoiding the low signal-to-noise issues that arise when relying solely on raw crime counts.
>
> **(2) Hierarchical hypergraph modeling with online adaptation.**
> Heterogeneous hyperedges are further organized into a homogeneous hypergraph to capture high-level co-occurrence and coupling among crime patterns. Online adaptation continually maintains these structures under distribution drift, enabling stable and reliable pattern extraction even when event data are scarce.
>
> **(3) Partially frozen LLM backbone.**
> The PF-LLM contributes strong pretrained reasoning priors for interpreting sparse real-world crime sequences. By freezing the FFN and only adapting the attention layers, the model remains lightweight while effectively aligning LLM knowledge with sparse, evolving crime signals.
>
> ---
>
> > **Q2: True zero rate is an important metric; however, it is overlooked in this paper.**
> >
>
> A2: We thank the reviewer for highlighting the importance of True Zero Rate (TZR). We have added TZR evaluation in **Sec. 5.1 (Page 7)** for the occurrence prediction comparison. As shown in Table 2, ST-HHOL achieves **0.736 TZR on the NYC dataset** and **0.663 on the CHI dataset**, outperforming all baselines and demonstrating state-of-the-art performance. This indicates that ST-HHOL not only accurately predicts crime occurrences but also **reliably identifies zero-occurrence regions**, which is particularly critical in sparse crime data scenarios.
>
> | TZR | SVM | ARIMA | DCRNN | STGCN | DeepCrime | GMAN | ST-SHN | OneNet | DLF | ST-HHOL(Ours) |
> | :---: | :---: | :---: | :---: | :---: | :---: | :---: | :---: | :---: | :---: | :---: |
> | CHI | 0.482 | 0.475 | 0.564 | 0.558 | 0.589 | 0.588 | 0.647 | 0.615 | 0.652 | 0.663 |
> | NYC | 0.603 | 0.579 | 0.637 | 0.675 | 0.676 | 0.669 | 0.714 | 0.673 | 0.688 | 0.736 |
>
>
> ---
>
> > **Q3:  Importance of e multi-source auxiliary data is not comprehensively discussed. Additionally, ablation study of the other two datasets should also be included since the importance of online learning should be shown.**
> >
>
> A3: (1) Crime data are extremely sparse and exhibit strong region–crime heterogeneity. Without auxiliary signals, the model struggles to distinguish meaningful patterns from noise. The incorporated multi-source data (POI, 311, weather, socioeconomic) provide complementary environmental and mobility-driven cues, enabling robust recovery of heterogeneous latent crime patterns and significantly reducing sparsity-induced bias.
>
> (2) We added ablation experiments on CHI, NYC, PHI and TOR datasets in **Sec. 5.2. Figure 4** in the revised PDF reports **average MAE** for each configuration. Removing online learning consistently degrades performance across all datasets, demonstrating its necessity under temporal concept drift. Similarly, other components (heterogeneous and hierarchical hypergraphs, PF-LLM) each contribute to robust performance, with ST-HHOL achieving the lowest MAE:
>
> | Average MAE | NYC | CHI | PHI | TOR |
> | --- | --- | --- | --- | --- |
> | w/o $G^e$ | 0.692 | 0.789 | 3.615 | 0.813 |
> | w/o $G^o$ | 0.681 | 0.772 | 3.530 | 0.795 |
> | w/o $E_T$ | 0.633 | 0.738 | 3.247 | 0.764 |
> | w/o PF-LLM | 0.641 | 0.759 | 3.336 | 0.769 |
> | w/o OL | 0.652 | 0.742 | 3.439 | 0.787 |
> | ST-HHOL | 0.617 | 0.728 | 2.937 | 0.755 |

---

> > ### Comment · Reviewer_3QXX · 2025-11-23
> >
> > Thanks authors for the prompt response and I think it solves many of my concerns. I will increase my score to 6.

---

> ### Author Response · Authors · 2025-11-24
> **Sincere Gratitude from Authors**
>
> Dear Reviewer 3QXX,
>
> Thank you very much for your feedback and for raising your score. We really appreciate your time, engagement, and constructive insights during the review.
>
> Best regards,
>
> Authors of submission 4693

---

### Official Review · Reviewer_9AaK · 2025-11-01

**Soundness:** 4
**Presentation:** 3
**Contribution:** 4
**Rating:** 8
**Confidence:** 4

**Summary:**

This paper proposes ST-HHOL, a spatial-temporal hierarchical hypergraph online learning framework for urban crime prediction. The framework captures dual specificity of crime patterns through hierarchical hypergraph convolutional networks, enhances spatial-temporal reasoning via partially frozen LLMs, and employs an iterative online learning strategy to address non-stationarity and concept drift, achieving state-of-the-art performance on three real-world datasets.

**Strengths:**

1. The introduction section accurately identifies multiple challenges in crime prediction, such as data sparsity and distribution drift. The proposed solutions to address these challenges are novel and well-motivated.

2. The "online learning" scenario presented in this work is more suitable for real-world applications, which enhances the practical value of the proposed framework.

3. The experimental evaluation is comprehensive, including multiple state-of-the-art baselines and validation of the model across multiple dimensions.

**Weaknesses:**

1. The proposed method exhibits considerable complexity in its design, which may lead to an excessive number of parameters. This could potentially affect the model's scalability and computational efficiency.

2. The description of how the data sparsity problem is addressed lacks clarity and detail.

**Questions:**

1. Could you provide a more clear and detailed explanation of how the data sparsity problem is resolved in your framework? Specifically, which components of ST-HHOL directly contribute to mitigating sparsity, and what mechanisms enable this?

2. While I understand the "online learning" scenario that the paper aims to address, I am uncertain whether the definition and usage of "online learning" in this context strictly aligns with the formal definition in machine learning literature. Could you clarify your interpretation and justify its appropriateness for this application?

---

> ### Author Response · Authors · 2025-11-17
> **Response to Reviewer 9AaK**
>
> Dear Review 9AaK,
>
> We sincerely appreciate the time and effort you have spent providing insightful feedback on our paper. We are honored that you recognized our hard work. We have carefully considered each of your comments and have addressed them one by one.
>
>
>
> > **Q1: The proposed method exhibits considerable complexity in its design, which may lead to an excessive number of parameters. This could potentially affect the model's scalability and computational efficiency.**
> >
>
> A1: While ST-HHOL integrates several components, its parameter size is dominated by the frozen PF-LLM backbone, which does not scale with the number of regions. The spatial hypergraph components contribute only 0.0048M parameters per region, resulting in minimal growth (47.05M→48.30M when increasing regions from 6→266). Empirically, ST-HHOL achieves 0.44s training / 0.09s inference per epoch, comparable to or more efficient than some excellent baselines.
>
> It is also worth noting that the proposed online hierarchical hypergraph—the core of our contribution—is a flexible pattern-learning module. Its effectiveness does not fundamentally depend on the PF-LLM. In resource-constrained scenarios, one could adopt a lighter-weight spatio-temporal backbone while still retaining the competitive advantage afforded by our novel crime pattern modeling.
>
> ---
>
> > **Q2: Could you provide a more clear and detailed explanation of how the data sparsity problem is resolved in your framework? Specifically, which components of ST-HHOL directly contribute to mitigating sparsity, and what mechanisms enable this?**
> >
>
> A2: ST-HHOL addresses data sparsity through three key components:
>
> **(1) Heterogeneous multi-source contextual integration.**
> Motivated by the inductive biases in crime generation, ST-HHOL anchors each region–crime pair to dense external drivers (POI, 311, weather, socioeconomic data) and integrates them with heterogeneous weights. This preserves crime-type–specific semantics and enriches sparse observations, avoiding the low signal-to-noise issues that arise when relying solely on raw crime counts.
>
> **(2) Hierarchical hypergraph modeling with online adaptation.**
> Heterogeneous hyperedges are further organized into a homogeneous hypergraph to capture high-level co-occurrence and coupling among crime patterns. Online adaptation continually maintains these structures under distribution drift, enabling stable and reliable pattern extraction even when event data are scarce.
>
> **(3) Partially frozen LLM backbone.**
> The PF-LLM contributes strong pretrained reasoning priors for interpreting sparse real-world crime sequences. By freezing the FFN and only adapting the attention layers, the model remains lightweight while effectively aligning LLM knowledge with sparse, evolving crime signals.
>
> ---
>
> > **Q3:  While I understand the "online learning" scenario that the paper aims to address, I am uncertain whether the definition and usage of "online learning" in this context strictly aligns with the formal definition in machine learning literature. Could you clarify your interpretation and justify its appropriateness for this application?**
> >
>
> A3:   We adopt the modern definition of continual (online) spatio-temporal forecasting, which extends classical online learning to **non-stationary, streaming spatio-temporal data**. In this setting(EAC [1] and STRAP [2]),“online learning” refers to continual adaptation to evolving data streams, rather than per-sample updates. Formally, for each temporal stage $ \tau $:
>
> $ f_{\theta(\tau)}^* $= $\arg\min_{\theta(\tau)}$ $E_{D_\tau \sim P(\tau)}$ $\left[ \mathcal{L}\big(f_{\theta(\tau)}(G_\tau, X_\tau), Y_\tau\big) \right] $
>
> where each $ D_\tau = (G_\tau, X_\tau, Y_\tau) $ represents streaming spatio-temporal graph data under the evolving distribution $ P(\tau) $.
>
> ST-HHOL aligns with this formulation and is motivated by the intrinsic non-stationarity of crime dynamics:
>
> (1) Non-stationary distributions: Crime patterns naturally violate $ P_{\text{train}}(X,Y) = P_{\text{test}}(X,Y) $; concept drift can be formalized as $ D_{KL}(P_t(Y|X)|P_{t+\tau}(Y|X)) \ge \delta $.
>
> (2) Streaming organization: Data arrive sequentially (daily/weekly), processed chronologically without future access.
>
> (3) Stage-wise incremental optimization: Fine-tuning and retraining phases optimize $ f_{\theta(\tau)}^* $ at each stage, enabling fast adaptation to short-term fluctuations while retaining long-term knowledge.
>
>
>
> [1] Chen W, et al. Expand and compress: Exploring tuning principles for continual spatio-temporal graph forecasting. ICLR, 2025.
>
> [2] Zhang H, et al. STRAP: Spatio-Temporal Pattern Retrieval for Out-of-Distribution Generalization. NeurIPS, 2025.

---

> > ### Comment · Reviewer_9AaK · 2025-11-22
> >
> > Thank you for the authors' response. Most of my concerns have been satisfactorily addressed. I will maintain my original score.

---

> > > ### Author Response · Authors · 2025-11-22
> > > **Sincere Gratitude from Authors**
> > >
> > > Dear Reviewer 9AaK,
> > >
> > >
> > > Thank you for taking the time and effort to provide valuable feedback. Once again, we really appreciate your guidance!
> > >
> > > Best regards,
> > >
> > >  Authors of submission 4693

---

### Official Review · Reviewer_GDKr · 2025-11-01

**Soundness:** 2
**Presentation:** 3
**Contribution:** 2
**Rating:** 2
**Confidence:** 4

**Summary:**

The authors have proposed a spatio-temporal online learning model for crime prediction. It is a mix of hierarchical hypergraphs to model context-crime interactions with a partially frozen GPT-2 module for temporal reasoning and make use of a staged online training strategy to adapt to concept drift. The method is evaluated on three public city-crime datasets and shows improvements over baselines.

**Strengths:**

- The authors have addressed a real-world spatio-temporal problem with clear motivations in the paper.
- The proposed work combines hypergraphs and online updates in a sensible pipeline.
- Good experimental effort across multiple cities and metrics.
- The ablations and sensitivity studies are good to see.
- Interpretability figures are a plus in understanding and analyzing the work.

**Weaknesses:**

- The novelty in this paper is a little incremental. Hypergraph + online updates + partial LLM freezing feels like an engineering solution proposed by combing current methods.
- The claims around “hierarchical hypergraph” are mostly architectural rearrangements or engineering solution, not a fundamentally very new formulation.
- The “partial GPT-2 freezing” choice appears heuristic and could be swapped for many LLM-augmented temporal modules without changing the story.
- The concept-drift handling is also not theoretically grounded.
- There is no clear insight into why this model generalizes better, beyond “more components.”
- The ethical context is thin, considering the sensitivity of predictive policing. Have more discussions on that end.

**Questions:**

- What are the modeling insight that separates this from prior hypergraph-based crime methods beyond stacking modules?
- Why is GPT-2 the right backbone rather than a standard temporal transformer or Time-LLM? Is the gain coming from pretraining or just extra capacity?
- How is fairness monitored? Crime data is biased; what prevents feedback loops?

---

> ### Author Response · Authors · 2025-11-17
> **The First Part of the Response to Reviewer GDKr**
>
> Dear Review GDKr,
>
> We greatly appreciate your hard work during the review process. We are also pleased that you recognized several key features of our work in the Summary and Strengths section: the clear motivation behind addressing a real-world spatiotemporal problem, the sensible method design, the comprehensive experimental evaluation, and interpretability analysis.  We believe several concerns stem from misunderstandings or missing clarifications in the current draft. Below, we respond to each point individually and provide additional evidence to address these issues.
>
>
>
> > **Q1: The novelty in this paper is a little incremental. Hypergraph + online updates + partial LLM freezing feels like an engineering solution proposed by combing current methods.**
> >
>
> A1:  Our method is not a simple combination of existing components. Our contribution lies in a new system-level architecture that introduces innovations at each layer to address the unique challenges of sparse and drift-prone crime data. Specifically:
> **(1) A dynamic heterogeneous latent-pattern hypergraph instead of a standard hierarchical hypergraph.**
> Prior hierarchical hypergraphs only stack homogeneous node layers. Our hierarchy is defined over latent crime patterns jointly inferred from crime data and multi-source environment factors, with heterogeneous hyperedges encoding their multi-factor composition. This enables dynamic pattern heterogeneity modeling and time-varying cross-pattern coupling, which existing hypergraph formulations cannot capture.
>
> **(2) A new level-specific drift adaptation mechanism.**
>
> Existing hypergraphs are static or uniformly updated. We design scale-aware temporal updates: micro-level patterns fine-tune every τ (fast drift), macro-level pattern composition updates every T (slow drift). This yields a hierarchically dynamic hypergraph, a structural capability not supported by prior work.
>
> **(3) A principled partially frozen LLM for sparse spatio-temporal reasoning.**
>
> Freezing FFN layers preserves pretrained priors (avoiding overfitting under extremely sparse counts), while unfreezing attention layers allows flexible temporal dependency adaptation. This stability–plasticity tradeoff is functionally distinct from naive full tuning or full freezing and is essential for pattern induction in sparse crime series.
>
> ---
>
> > **Q2: The claims around “hierarchical hypergraph” are mostly architectural rearrangements or engineering solution, not a fundamentally very new formulation.**
> >
>
> A2:  We would like to clarify that our hierarchical hypergraph is not an architectural rearrangement of known components, but a **new formulation tailored for modeling heterogeneous and drifting crime patterns**. Our contributions differ from existing hypergraph structures in two fundamental aspects:
>
> **(1) A two-level latent-pattern hypergraph that merges heterogeneous pattern formation with homogeneous pattern coupling.**
>
> Prior hypergraphs directly operate on raw nodes (regions and categories). In contrast, we first construct **latent crime patterns** via _heterogeneous hyperedges_ that jointly encode multi-source environmental factors and crime signals—capturing high-order, co-existing, and region-specific influences that cannot be extracted from raw sparse data. We then treat these patterns as **hypernodes** and use a _homogeneous hypergraph_ to model their **time-varying co-occurrence dynamics**.
> This two-stage formulation (heterogeneous pattern induction → homogeneous pattern coupling) is **absent in prior hierarchical hypergraphs**, which typically only stack structural layers.
>
> **(2) A level-specific temporal adaptation mechanism reflecting real-world drift.**
>
> Environmental factors drift slowly, whereas crime co-occurrence relationships fluctuate rapidly. We encode this asymmetry by assigning **different update frequencies** to heterogeneous and homogeneous hyperedges, enabling **hierarchically adaptive drift modeling**. Existing hypergraph designs do not support such level-specific temporal dynamics, which are essential for streaming crime data.

---

> ### Author Response · Authors · 2025-11-17
> **The Second Part of the Response to Reviewer GDKr**
>
> > **Q3：The “partial GPT-2 freezing” choice appears heuristic and could be swapped for many LLM-augmented temporal modules without changing the story.**
> >
>
> A3:  Our choice of partially frozen GPT-2 is principled rather than heuristic.
> (1) We adopt **GPT-2 Small (117M)** as a balanced open-source LLM that offers strong linguistic priors with affordable computational cost, enabling frequent online adaptation in resource-constrained spatio-temporal settings.
> (2) The **partial freezing strategy** is specifically designed to **transfer pretrained linguistic reasoning into sparse crime sequences while stabilizing training and preventing overfitting**. This effectively enhances model robustness by leveraging stable linguistic knowledge as an anchor, which is crucial under data sparsity, as verified by our ablation results.
> Our goal is not to benchmark LLM variants but to **bridge LLM reasoning with spatio-temporal learning**—leveraging pretrained priors to mitigate sparsity while maintaining adaptability and efficiency. The paper's core novelty lies in the hierarchical hypergraph and online adaptation design, while the LLM component serves as a theoretically motivated and empirically validated reasoning enhancer.
>
> ---
>
> > **Q4: The concept-drift handling is also not theoretically grounded.**
> >
>
> A4:   Our drift-handling mechanism is theoretically grounded  in two aspects:
>
> (1)  We explicitly formulate drift as temporal variation in the joint distribution  $ P_t(X,Y) $ in Sec 4.3 and model it through _time-evolving heterogeneous and homogeneous hypergraphs_ that separately capture **slow structural drift** (environment-driven pattern shifts) and **fast relational drift** (crime co-occurrence dynamics).
> (2) Our update schedule is **data-driven rather than heuristic**. FFT-based analysis on three datasets reveals strong weekly–monthly periodicities, time adjustment analysis also shows that bi-weekly fine-tuning and bi-monthly retraining align with empirical dominant frequencies (Sec. 5.3). Thus, both the _formulation_ and the _adaptation schedule_ are theoretically justified.
>
> | Dataset | CHI | NYC | PHI |
> | :---: | :---: | :---: | :---: |
> | Q1 | 7 | 6.25 | 9 |
> | Q2 | 22.25 | 24 | 31 |
> | Q3 | 65.5 | 66.5 | 65.5 |
>
> ---
>
> > **Q5: There is no clear insight into why this model generalizes better, beyond “more components.”**
> >
>
> A5:  Our generalization improvement arises from structured inductive bias and data-adaptive learning. The hierarchical heterogeneous hypergraph **aligns with** **the true generative process of spatio-temporal crimes**—capturing factor-specific heterogeneity and cross-pattern coupling.   The online learning strategy is designed for **multi-period drift**, preventing catastrophic forgetting caused by frequent updates while retaining sufficient plasticity to adapt to new crime patterns.  Moreover,   PF-LLM freezes the FFN to retain few-shot reasoning priors learned from text, while updating attention layers to adapt to spatio-temporal dependencies in sparse crime data, effectively **transferring LLM reasoning to the prediction task**.
>
> Ablation results (Sec. 5.2) consistently show that removing any component leads to clear drops in performance, confirming that the gains come from principled architectural choices rather than component accumulation.
>
> ---
>
> > **Q6: The ethical context is thin, considering the sensitivity of predictive policing. Have more discussions on that end.**
> >
>
> A6:  We have expanded the discussion on the ethics statement in **Sec. 8**. ST-HHOL is designed for macro-level analysis of spatio-temporal crime trends, academic research, and public safety planning, and is explicitly not intended for tactical decisions or individual-level risk assessment. By focusing on aggregated crime patterns rather than individual-level predictions, and by excluding sensitive features such as demographic or police station data, the model inherently limits risks of unfair targeting or over-policing. Empirical evaluation shows stable performance across regions with diverse socioeconomic conditions, indicating a low likelihood of harmful “bad cases.” These design choices and validations ensure that ST-HHOL serves as a descriptive, analytical tool, mitigating ethical risks while enabling insights into heterogeneous crime dynamics.

---

> ### Author Response · Authors · 2025-11-17
> **The Third Part of the Response to Reviewer GDKr**
>
> > **Q7: What are the modeling insight that separates this from prior hypergraph-based crime methods beyond stacking modules?**
> >
>
> A7:  Our approach is designed from first principles to capture both the **heterogeneity** and **dynamic coupling of latent crime patterns**. Existing methods construct homogeneous hyperedges over sparse $ T \times N \times C $ crime tensors, failing to model pattern-level heterogeneity shaped by multi-source environmental factors, their dynamic co-occurrence, or level-specific adaptation to concept drift.
>
> We introduce two key innovations:
>
> **（1）Dynamic modeling of heterogeneous latent patterns.** Each region–crime pair is treated as an anchor, and heterogeneous hyperedges encode factor-specific influences. This mitigates sparsity and models time-varying co-occurrence among latent patterns—beyond the reach of homogeneous or flat hierarchical hypergraphs.
>
> **（2）Level-specific temporal adaptation.** Heterogeneous hyperedges are elevated into higher-level hypernodes, forming a hierarchy. Micro-level patterns (high-frequency fluctuations) are fine-tuned every $ \tau
>  $ steps, while macro-level compositions (slowly drifting influences) update every $ T $ steps, enabling a hierarchically dynamic hypergraph with both stability and temporal responsiveness.
>
> ---
>
> > **Q8: Why is GPT-2 the right backbone rather than a standard temporal transformer or Time-LLM? Is the gain coming from pretraining or just extra capacity?**
> >
>
> A8:  We use GPT-2 (Small) to transfer the few-shot reasoning capabilities learned from text pretraining to modeling sparse spatio-temporal crime dependencies. Standard Transformers lack such pretrained knowledge, as confirmed by our w/o PF-LLM ablation study results (Sec 5.2, page 8, Fig 4). Larger LLMs (such as Time-LLM) could improve absolute performance, but our focus is on lightweight, partially frozen LLMs achieving effective reasoning on real-world crime data. GPT-2 Small balances performance and efficiency, fulfilling this goal.
>
> ---
>
> > **Q9: How is fairness monitored? Crime data is biased; what prevents feedback loops?**
> >
>
> A9:  Spatial skewed bias in crime data primarily arises from sparsity. To mitigate this, our framework incorporates two mechanisms:
>
> (1) **Technical mitigation:** The heterogeneous hypergraph integrates multi-source contextual factors—such as POI, 311 calls, weather, and socioeconomic data—to uncover latent crime patterns and heterogeneous semantics, rather than relying solely on sparse raw crime data. Meanwhile, PF-LLM transfers pretrained reasoning capabilities to improve temporal modeling in low-signal scenarios.
>
> (2) **Evaluation mitigation:** We provide both **quantity** and **occurrence** predictions, separating event intensity from occurrence frequency. This ensures fair representation of low- and high-crime regions and prevents reinforcing skewed patterns. Empirically, as validated in Appendix E.2 (Fig. 9), our model demonstrates robust performance across both low- and high-crime regions, showing no signs of systematic bias amplification.

---

> > ### Comment · Reviewer_GDKr · 2025-11-17
> >
> > Thank you for the clarification. I have updated my score.

---

> > > ### Author Response · Authors · 2025-11-17
> > > **Thank you for your time and consideration**
> > >
> > > Dear Reviewer GDKr,
> > >
> > > We are glad to hear that our rebuttal effectively addressed your concerns. Thank you again for taking the time and effort to provide valuable feedback on our paper.
> > >
> > > Best wishes,
> > > Authors of submission 4693

---

### Official Review · Reviewer_pemE · 2025-11-03

**Soundness:** 3
**Presentation:** 4
**Contribution:** 2
**Rating:** 6
**Confidence:** 4

**Summary:**

This paper addresses urban crime prediction by tackling two key challenges: (1) sparse crime records that fail to capture latent high-order patterns shaped by heterogeneous contextual factors, and (2) high non-stationarity that renders conventional offline models ineffective against concept drift. The authors propose ST-HHOL, a Spatio-Temporal Hierarchical Hypergraph Online Learning framework that integrates a hierarchical hypergraph convolution network with a Partially-Frozen LLM (PF-LLM) and an iterative online learning strategy. Experimental results on three real-world datasets demonstrate that ST-HHOL consistently outperforms state-of-the-art methods across various metrics while providing enhanced interpretability.

**Strengths:**

1. **Well-motivated problem formulation**: The paper is well-motivated to solve the sparse data problem and the concept drift challenge in crime prediction. Particularly noteworthy is the authors' approach to addressing concept drift through an innovative online learning mechanism that explicitly disentangles spatially invariant and temporally variant components. This separation allows the model to freeze spatial parameters (Θs) associated with crime patterns while adapting temporal parameters (Θd(t)) for co-occurrence relationships, providing a principled solution to handle both short-term fluctuations and long-term distributional shifts.

2. **Novel online learning mechanism**: The paper proposes a sophisticated iterative two-phase update scheme that combines frequent fine-tuning (every τ steps) for rapid adaptation to recent fluctuations with periodic retraining (every T steps) for capturing long-term gradual shifts. This mechanism effectively addresses the non-stationary nature of crime dynamics by balancing short-term responsiveness and long-term stability, which is a significant contribution to the online learning literature in spatio-temporal forecasting.

3. **Effective LLM integration strategy**: Experiments demonstrate that the Partially-Frozen LLM (PF-FFN) approach is effective in leveraging pre-trained sequence modeling priors while adapting to crime-specific dependencies. The ablation study shows that this approach achieves a better trade-off between retaining pretrained knowledge and adapting to the target domain compared to fully frozen or fully tuned alternatives.

4. **Comprehensive experimental evaluation**: The experimental evaluation is thorough and well-designed, featuring 14 diverse baselines spanning traditional methods (SVM, ARIMA), spatio-temporal forecasting models (DCRNN, STGCN, AGCRN, MTGNN, GMAN), crime-specific models (DeepCrime, ST-HSL, ST-SHN), and online learning models (DLF, FSNet, OneNet). Beyond performance comparison, the authors conduct extensive analyses including robustness studies, scalability analysis on NYC Taxi data, efficiency analysis, hyperparameter sensitivity studies, and interpretability case studies with visualizations.

**Weaknesses:**

1. **Limited novelty in hierarchical hypergraph design**: Though having some differences in motivation, hierarchical hypergraph neural networks and various GNN or hyperGNN designs have been fairly explored in previous works one to two years ago. The key innovation and contribution of this hierarchical hypergraph construction (heterogeneous G^e for crime patterns and homogeneous G^o for co-occurrence) is not very prominent considering the existing literature on hypergraph-based spatial-temporal modeling.

2. **Incomplete ablation study for LLM component**: The paper lacks a complete ablation experiment that entirely removes the PF-LLM component. While the authors compare different LLM variants (FPT, No Pretrain, Full Tuning, PF-A, PF-FFN), there is no comparison with a baseline that replaces the LLM with traditional neural networks (e.g., RNN, CNN, or MLP). This makes it difficult to assess the true necessity and contribution of the LLM component to the overall performance.

3. **Limited LLM baseline consideration**: The paper uses GPT-2, which has relatively low foundational performance compared to existing near-human-level language models. This raises questions about the necessity of fine-tuning such a weak model. It would be valuable to explore whether more powerful large models could achieve better results through prompt-based in-context learning rather than parameter fine-tuning, especially given the potential computational overhead of the current approach.

**Questions:**

1. Given that hierarchical hypergraph approaches have been extensively studied in recent years, what specific architectural innovations or theoretical contributions distinguish your hierarchical hypergraph design from existing methods beyond the particular application to crime prediction?

2. Could you provide an ablation study that completely removes the PF-LLM component and replaces it with traditional sequence modeling approaches (e.g., LSTM, GRU, or Transformer without pre-training)? This would help quantify the actual contribution of the LLM component.

3. Have you considered experimenting with more recent and powerful language models? Would a comparison with prompt-based approaches using stronger LLMs (e.g., through in-context learning) provide insights into whether the fine-tuning approach is truly necessary, or if the computational cost could be better justified?

---

> ### Author Response · Authors · 2025-11-17
> **Response to Reviewer pemE**
>
> Dear Review pemE,
>
> We sincerely thank you for your hard work on our paper. We carefully considered each of your comments and dealt with them accordingly.
>
>
>
> > **Q1: Given that hierarchical hypergraph approaches have been extensively studied in recent years, what specific architectural innovations or theoretical contributions distinguish your hierarchical hypergraph design from existing methods beyond the particular application to crime prediction.**
> >
>
> A1: Our design introduces two methodological innovations :
>
> **(1) Dynamic modeling of heterogeneous latent patterns.**
> Unlike prior hierarchical hypergraphs that organize homogeneous nodes into structural layers, our hierarchy is defined over **latent crime patterns**, inferred from both crime records and multi-source environmental factors. Heterogeneous hyperedges encode the multi-factor contributions to each pattern, enabling **dynamic modeling of pattern-level heterogeneity and time-varying co-occurrence**, which cannot be captured by existing hypergraph architectures.
>
> **(2) Level-specific temporal adaptation.**
> Existing hierarchical hypergraphs are either static or uniformly updated. We introduce **scale-aware temporal updates**: micro-level latent patterns (high-frequency fluctuations) are fine-tuned every τ steps, while macro-level pattern composition (slowly drifting environmental influences) is updated every T steps. This yields a **hierarchically dynamic hypergraph** with level-specific drift adaptation—an architectural feature absent in prior work.
>
> ---
>
> > **Q2: Could you provide an ablation study that completely removes the PF-LLM component and replaces it with traditional sequence modeling approaches (e.g., LSTM, GRU, or Transformer without pre-training)? This would help quantify the actual contribution of the LLM component.**
> >
>
> A2:  We conducted an ablation study by replacing PF-LLM with a standard Transformer on all four crime datasets (CHI, NYC, PHI, TOR). The results are shown in the Table below:
>
> | Variants | CHI | NYC | PHI | TOR |
> | --- | --- | --- | --- | --- |
> | w/o PF-LLM | 0.641 |0.759 | 3.336 | 0.769 |
> | ST-HHOL | 0.617 |0.728 | 2.937 | 0.755 |
>
>
>  PF-LLM consistently improves performance across datasets, reducing prediction error by 3–12%. This confirms that **pre-trained sequence modeling knowledge from the LLM is effectively transferred to sparse crime data**, enabling more accurate reasoning over low-density events.
> We have updated **Sec. 5.2** to include these results and present all ablation experiments across datasets (see Figure 4).
>
> ---
>
> > **Q3: Have you considered experimenting with more recent and powerful language models? Would a comparison with prompt-based approaches using stronger LLMs (e.g., through in-context learning) provide insights into whether the fine-tuning approach is truly necessary, or if the computational cost could be better justified?**
> >
>
> A3:  Thank you for the valuable suggestion. While stronger and larger LLMs could potentially improve absolute performance, exploring larger models is not the goal of this work. Our use of a **partially frozen GPT-2 (Small)** is a deliberate design choice aimed at transferring general LLM reasoning capabilities to the spatio-temporal dependency modeling of sparse crime data in a resource-efficient manner. We find that GPT-2 provides sufficient capacity for our task, and increasing the model scale would not better serve the core objective of this study.

---

> ### Author Response · Authors · 2025-11-28
> **Kindly Request for Reviewer's Feedback**
>
> Dear Reviewer pemE,
>
> **Since the End of the Rebuttal is coming very soon - only a few days left**, we would like to inquire if our response addresses your primary concerns. If you have any additional suggestions, we are more than willing to engage in further discussions and make necessary improvements to the paper.
>
> Thanks again for dedicating your time to enhancing our paper!
>
> Looking forward to your feedback.
>
> Best,
>
> Authors of submission 4693

---

### Official Review · Reviewer_dQeU · 2025-11-08

**Soundness:** 2
**Presentation:** 3
**Contribution:** 2
**Rating:** 4
**Confidence:** 3

**Summary:**

This paper proposes a ST-HHOL for crime prediction to tackle the insufficiency of variety, high-order dependency, and non-stationality in sparse crime records. ST-HHOL includes (i) hypergraph modeling for integrating crime data with heterogeneous contextual factors and their co-occurrence relations, (ii) an iterative online learning for addressing concept drift by employing frequent fine-tuning for short-term dynamics and periodic retraining for long-term shifts, and (iii) a partially-frozen LLM for enhancing spatio-temporal reasoning under sparse supervision. The authors conducted extensive experiments on three real-world datasets and demonstrated that ST-HHOL consistently outperforms state-of-the-art methods in terms of accuracy and robustness, while also providing enhanced interpretability. In summary, this paper proposes an well-motivated designs for crime prediction to address the problems, conducts extensive experiments, and provides reasonable readability. However, this study would be further improved: (i) more theoretical justification on, for example, regret analysis of online learning and criteria for determining timings of fine-tuning and retraining; (ii) scalability and computational complexity of ST-HHOL; (iii) evaluation of concept drift (what types of concept drifts and how ST-HHOL adapt to them); (iv) empirical demonstration of ST-HHOL on datasets outside U.S.

**Strengths:**

- Well-motivated design: To address the challenges in crime prediction, this paper unifies three components: hypergraph modeling, online learning, and pretrained sequence prediction.
- Extensive experiments: The authors conducted extensive experiments on many cities, ablation study, hyperparameter study, and visualization.
- Readability: The paper is overall easy to follow

**Weaknesses:**

- Theoretical background: Most of the proposed algorithm are based on intuitive and empirical insights. In other words, the theoretical background is relatively weak. For example, regret analysis on online learning (Shalev-Shwartz 2012) and criteria for determining $\tau$ (fine-tuning) and $T$ (retraining) are missing.
- Scalability: Hypergraph construction and periodic retraining may become expensive for large-scale city meshes. The computational complexity of ST-HHOL and its empirical results for larger datasets should be discussed.
- Evaluation under Concept Drift: The experimental results do not provide what concept drift types are in the datasets (e.g., sudden, gradual, reccuring) (Gama et al. 2014) and how ST-HHOL adapt to such drifts.
- Variety of Datasets: This paper conducts experiments on US cities datasets only, and the numbers of regions are up to order 100.

[Shalev-Shwartz 2012] Shai Shalev-Shwartz: Online learning and online convex optimization. Foundations and Trends® in Machine Learning, 4(2), pp.107-194, 2012.

[Gama 2014] Joao Gama, Indre Zliobaite, Albert Bifet, Mykola Pechenizkiy, and Abdelhamid Bouchachia: A survey on concept drift adaptation. ACM Computing Surveys, 64(4), pp.1-37, 2014.

**Questions:**

Please answer the points described in the Weaknesses.

---

> ### Author Response · Authors · 2025-11-17
> **The First Part of the Response to Reviewer dQeU**
>
> Dear Reviewer  dQeU,
>
> We greatly appreciate your hard work during the review process. We will address your concerns one by one to clarify any confusion.
>
> ****
>
> > **Q1: Theoretical background: Most of the proposed algorithm are based on intuitive and empirical insights. In other words, the theoretical background is relatively weak. For example, regret analysis on online learning (Shalev-Shwartz 2012) and criteria for determining  (fine-tuning) and  (retraining) are missing.**
> >
>
> A1:  Our work does not target theoretical online learning, the online setting is introduced solely to handle concept drift in real-world crime prediction. Therefore, classical regret-style analysis is not directly applicable, as our focus is practical sequential forecasting rather than worst-case optimization guarantees.
>
> Our update strategy is structurally motivated. Long-term, slowly evolving trends are encoded in the heterogeneous hypergraph, which is updated less frequently, while short-term, volatile co-occurrence patterns are captured by the homogeneous hypergraph, which is updated more often. This asymmetric update design reflects the multi-scale nature of crime data and avoids unnecessary full retraining.
>
> To select update intervals, we rely on data-driven evidence: **FFT** decomposition consistently reveals (i) a dominant weekly component, (ii) a 3–4 week median period, and (iii) a lower-quartile period of about two months. These empirical patterns support our week-level fine-tuning and month-level retraining choices. **We have added the basis of update frequency in Sec 5.3 (Page 8).**
>
> | Dataset | CHI | NYC | PHI |
> | :---: | :---: | :---: | :---: |
> | Q1 | 7 | 6.25 | 9 |
> | Q2 | 22.25 | 24 | 31 |
> | Q3 | 65.5 | 66.5 | 65.5 |
>
>
> ---
> > **Q2: Scalability: Hypergraph construction and periodic retraining may become expensive for large-scale city meshes. The computational complexity of ST-HHOL and its empirical results for larger datasets should be discussed.**
> >
>
> A2：The hypergraph construction in ST-HHOL is lightweight: both heterogeneous and homogeneous hypergraphs are built from aggregated region–crime statistics in **O(|V| + |E|)** without enumerating high-order combinations, so their size grows nearly linearly with the number of regions. Periodic fine-tuning and retraining are also scalable, as the update cost is dominated by the PF-LLM backbone, while the hypergraph layers contribute only a small fraction of the parameters.
>
> Empirically, scaling from small cities (PHI: 6 regions) to larger ones (TOR: 158, NYC Taxi: 266), the model size increases only marginally (47.0M → 48.3M), and the retraining time remains stable (<5% variation). These results confirm that ST-HHOL maintains efficiency even on large-scale city meshes.
>
> | Datesets | PHI | CHI | NYC | TOR | NYC taxi |
> | :---: | :---: | :---: | :---: | :---: | :---: |
> | Regions | 6 | 77 | 123 | 158 | 266 |
> | Parameters (M) | 47.050 | 47.391 | 47.611 | 47.780 | 48.299 |
> ---
> > **Q3: Evaluation under Concept Drift: The experimental results do not provide what concept drift types are in the datasets (e.g., sudden, gradual, reccuring) (Gama et al. 2014) and how ST-HHOL adapt to such drifts.**
> >
>
> A3: In real-world crime prediction, the underlying dynamics exhibit multi-scale temporal coupling—daily rhythms, weekend effects, seasonal patterns, and irregular, sparse bursts—making the concept drift inherently mixed rather than belonging to a single canonical type. For this reason, our method is _not designed for any specific drift type_, but instead provides a flexible adaptation mechanism.
>
> In ST-HHOL, the update frequency $ T $ and $ \tau $ allow the model to adjust to different drift behaviors: denser updates can handle sudden shifts, while slower schedules align with gradual or periodic drifts. This tunable mechanism enables the model to cope with heterogeneous and evolving drift patterns without requiring prior assumptions about the drift category.
>
> ---

---

> ### Author Response · Authors · 2025-11-17
> **The Second Part of the Response to Reviewer dQeU**
>
> > **Q4: Variety of Datasets: This paper conducts experiments on US cities datasets only, and the numbers of regions are up to order 100.**
> >
>
> A4：We thank the reviewer for this suggestion. To broaden geographic diversity, **we added a new dataset from Toronto, Canada**, resulting in four crime datasets covering two countries and cities of different sizes (6–**158 regions**). Publicly available crime data are typically aggregated at the region or precinct level, and prior work[1,2,3] in this field also relies on NYC and CHI for this reason; the spatial granularity of 100+ regions already represents the upper limit of current open-data crime repositories.
>
> | Datasets | Region Num | Time Span | Time Interval | Crime Data |
> |----------|------------|-----------|---------------|------------|
> | CHI      | 77         | January 1, 2023 – December 31, 2024 | 1 day | Theft, Battery, Assault, Damage |
> | NYC      | 123        | January 1, 2022 – December 31, 2024 | 1 day | Larceny, Assault, Mischief, Robbery |
> | PHI      | 6          | January 1, 2023 – December 31, 2024 | 1 day | Theft, Assault, Vehicle, Mischief |
> | TOR      | 158        | January 1, 2023 – December 31, 2024 | 1 day | Assault, B&E, Robbery, Theft |
>
>
> To further assess scalability beyond crime data, we additionally evaluate ST-HHOL on the **NYC Taxi dataset** with **266 sensors**, which is substantially larger than any crime dataset. ST-HHOL maintains competitive performance on this larger spatial mesh, demonstrating that the method generalizes well across cities and scales.
>
> The **complete experimental results on the newest dataset** (TOR) are included  **in Sec. 5.1 (p. 7)** and **Sec. E.1–E.2 (pp. 19–20)**.
>
> [1]Liang K, et al. Hawkes-enhanced spatial-temporal hypergraph contrastive learning based on criminal correlations. AAAI, 2024.
>
> [2] Wu Z, et al. Spatial-Temporal Mixture-of-Graph-Experts for Multi-Type Crime Prediction. arXiv, 2024.
>
> [3] Li Z, et al. Spatial-temporal hypergraph self-supervised learning for crime prediction. ICDE, 2022.

---

> ### Author Response · Authors · 2025-11-17
> **The Third Part of the Response to Reviewer dQeU**
>
> **Supplement to Q4**:
>
> The complete experiment results on the newest dataset TOR are shown in the Table below. For quantity prediction, it reduces average MAE and MAPE by 4.12\% and 7.94\% on TOR, demonstrating its effectiveness and generalizability.
>
> **Table 10: Overall performance of crime prediction on TOR dataset. The results are 5-run error comparison, the bold font indicates the best result.**
>
> | Method | Assault MAE | Assault RMSE | Assault MAPE | B&E MAE | B&E RMSE | B&E MAPE | Robbery MAE | Robbery RMSE | Robbery MAPE | Theft MAE | Theft RMSE | Theft MAPE |
> |--------|-------------|--------------|--------------|---------|----------|----------|-------------|--------------|--------------|-----------|-----------|-----------|
> | SVM | 0.74 | 1.17 | 0.53 | 1.62 | 2.04 | 0.56 | 0.83 | 1.15 | 0.67 | 1.54 | 1.77 | 0.54 |
> | ARIMA | 0.71 | 1.18 | 0.50 | 1.56 | 2.02 | 0.55 | 0.89 | 1.16 | 0.68 | 1.52 | 1.73 | 0.53 |
> | DCRNN | 0.74 ±0.05 | 1.12 ±0.10 | 0.44 ±0.03 | 1.50 ±0.10 | 2.50 ±0.20 | 0.45 ±0.05 | 0.63 ±0.07 | 1.14 ±0.18 | 0.20 ±0.03 | 2.13 ±0.16 | 3.07 ±0.24 | 0.47 ±0.05 |
> | STGCN | 0.83 ±0.02 | 1.26 ±0.05 | 0.49 ±0.04 | 1.43 ±0.06 | 2.41 ±0.16 | 0.43 ±0.06 | 0.71 ±0.22 | 1.15 ±0.27 | 0.21 ±0.01 | 2.04 ±0.44 | 2.96 ±0.52 | 0.50 ±0.05 |
> | AGCRN | 0.62 ±0.01 | 0.97 ±0.06 | 0.35 ±0.01 | 0.98 ±0.10 | 1.72 ±0.35 | 0.43 ±0.05 | 0.52 ±0.08 | 1.12 ±0.09 | 0.30 ±0.07 | 1.01 ±0.09 | 1.70 ±0.35 | 0.50 ±0.07 |
> | MTGNN | 0.93 ±0.04 | 1.32 ±0.07 | 0.56 ±0.08 | 1.45 ±0.34 | 2.30 ±0.56 | 0.58 ±0.18 | 0.86 ±0.12 | 1.28 ±0.16 | 0.48 ±0.05 | 1.45 ±0.34 | 2.34 ±0.42 | 0.51 ±0.17 |
> | GMAN | 0.98 ±0.04 | 1.37 ±0.05 | 0.52 ±0.05 | 1.30 ±0.07 | 2.20 ±0.10 | 0.55 ±0.06 | 1.31 ±0.01 | 1.77 ±0.05 | 0.49 ±0.03 | 1.30 ±0.03 | 2.21 ±0.12 | 0.59 ±0.03 |
> | MoSSL | 0.74 ±0.02 | 0.93 ±0.01 | 0.41 ±0.01 | 1.02 ±0.04 | 1.71 ±0.02 | 0.40 ±0.01 | 0.54 ±0.01 | 1.06 ±0.15 | 0.21 ±0.03 | 1.08 ±0.05 | 1.70 ±0.01 | 0.45 ±0.07 |
> | DeepCrime | 0.70 ±0.05 | 1.16 ±0.06 | 0.41 ±0.02 | 1.33 ±0.05 | 2.09 ±0.15 | 0.49 ±0.08 | 0.68 ±0.15 | 1.32 ±0.06 | 0.40 ±0.07 | 1.27 ±0.18 | 2.14 ±0.35 | 0.56 ±0.10 |
> | ST-SHN | 0.72 ±0.03 | 1.13 ±0.03 | 0.35 ±0.06 | 1.14 ±0.02 | 1.82 ±0.05 | 0.47 ±0.08 | 0.57 ±0.12 | 1.12 ±0.05 | 0.36 ±0.05 | 1.21 ±0.05 | 1.90 ±0.27 | 0.56 ±0.08 |
> | ST-HSL | 0.64 ±0.10 | 1.02 ±0.05 | 0.42 ±0.06 | 1.08 ±0.15 | 2.93 ±0.01 | 0.45 ±0.03 | 0.79 ±0.05 | 1.18 ±0.05 | 0.47 ±0.03 | 1.14 ±0.01 | 1.99 ±0.01 | 0.54 ±0.02 |
> | DLF | 1.05 ±0.04 | 1.76 ±0.05 | 0.33 ±0.14 | 1.94 ±0.06 | 3.01 ±0.10 | 0.49 ±0.03 | 1.02 ±0.15 | 2.03 ±0.20 | 0.30 ±0.20 | 1.76 ±0.15 | 3.11 ±0.14 | 0.50 ±0.03 |
> | FSNet | 0.78 ±0.13 | 1.63 ±0.20 | 0.39 ±0.08 | 1.38 ±0.16 | 2.73 ±0.24 | 0.47 ±0.03 | 1.12 ±0.08 | 1.60 ±0.09 | 0.32 ±0.07 | 1.46 ±0.13 | 3.04 ±0.36 | 0.54 ±0.11 |
> | OneNet | 0.94 ±0.06 | 1.91 ±0.14 | 0.43 ±0.08 | 1.46 ±0.17 | 2.92 ±0.34 | 0.48 ±0.04 | 1.09 ±0.07 | 1.24 ±0.10 | 0.35 ±0.08 | 1.52 ±0.16 | 2.75 ±0.41 | 0.52 ±0.04 |
> | **ST-HHOL (Ours)** | **0.58 ±0.01** | **0.89 ±0.01** | **0.31 ±0.01** | **0.96 ±0.01** | **1.68 ±0.01** | **0.39 ±0.01** | **0.50 ±0.04** | **1.02 ±0.02** | **0.18 ±0.01** | **0.98 ±0.01** | **1.70 ±0.01** | **0.40 ±0.01** |

---

> ### Comment · Reviewer_dQeU · 2025-11-20
> **Official comment by Reviewer dQeU**
>
> Thank you for your clarification. I am satisfied with all your responses.

---

> > ### Author Response · Authors · 2025-11-21
> > **Sincere Gratitude from Authors**
> >
> > We sincerely appreciate your thoughtful review and constructive feedback. We are glad that our revisions and clarifications have addressed your concerns, and we thank you once again for your time and valuable insights.
> >
> > Best wishes,
> >
> > Authors of submission 4693

---

### Author Response · Authors · 2025-11-18
**General Response to All Reviewers (the summary of revisions)**

We would like to express our gratitude to Reviewers dQeU, pemE, GDKr, 9AaK, and 3QXX for their diligent review and constructive feedback. In accordance with the modifications requested by the reviewers, we have revised the manuscript as follows:

+ **Section 2 (Related Work):** We expanded the comparison with existing hypergraph-based crime prediction methods, emphasizing our novelty in modeling _heterogeneous latent crime patterns_, _capturing their dynamic coupling_, and _introducing hierarchical, multi-scale concept-drift adaptation_, thereby addressing reviewers' concerns (pemE Q1; GDKr Q1–Q2, Q7).
+ **Sections 4.1–4.2 (Methodology):** We clarified how the proposed hierarchical hypergraph and PF-LLM jointly _mitigate data sparsity_ and stabilize spatio-temporal dependency modeling, addressing reviewers' questions (GDKr Q9; 9AaK Q2; 3QXX Q1).
+ **Section 5.1 (Main Results):** We added a new Canadian crime dataset (Toronto, 158 regions) for quantity prediction to _enhance dataset diversity_, with full results provided in Appendix E.1. We also included the _True Zero Rate (TZR)_ metric for occurrence prediction, responding to reviewers' comments (dQeU Q4; 3QXX Q2).
+ **Section 5.2 (Ablation Study):** We provided complete _ablation results across all four datasets_, including experiments w/o PF-LLM (replaced by a standard Transformer). These results consistently validate the contribution of each component in ST-HHOL, addressing reviewers' concerns (pemE Q2; GDKr Q8; 3QXX Q3).
+ **Section 5.3 (Time Adjustment Analysis):**  We clarified the _theoretical rationale for choosing the update frequencies_ $ T $ and $ \tau $, supported by _FFT-based analysis_ that reveals weekly and monthly fluctuation patterns in crime data, thereby addressing reviewers' questions (dQeU Q1; GDKr Q4).
+ **Section 8 (Ethics Statement):** We strengthened the ethical discussion, clarifying the intended research-oriented use of the model and demonstrating that the predictions do not exhibit harmful failure cases, responding to reviewers' concerns (GDKr Q6).

All revisions are highlighted in **blue** in the  revised PDF.  We hope these revisions adequately address the reviewers' concerns and are grateful once again for their constructive suggestions.

Best regards,

All Authors of Submission 4693.

---

### Author Response · Authors · 2025-12-01
**Author Final Remarks To AC and All Reviewers**

Dear Area Chair, Reviewers dQeU, pemE, GDKr, 9AaK, and 3QXX,

We sincerely thank the Area Chair and all reviewers for the time, effort, and constructive feedback provided throughout the evaluation process, especially under the unexpected circumstances. We are writing to provide a summary of our rebuttal progress to assist in your evaluation.

As of **November 24**, our submission has achieved complete positive scores of **6, 6, 6, 8, 6**. We are encouraged that the reviewers recognized multiple strengths of our submission, including `the importance and practical relevance of the research problem` (Reviewers 9AaK, GDKr, pemE), `the well-motivated and novel method design` (Reviewers dQeU, pemE, 9AaK, 3QXX), `the comprehensive experimental evaluation` (Reviewers dQeU, pemE, GDKr, 9AaK, 3QXX), `the strong empirical performance` (Reviewers dQeU, pemE, 9AaK, 3QXX), and `the clarity of the presentation` (Reviewers dQeU, GDKr, 3QXX).

------

We have provided comprehensive, point-by-point responses to all reviewer questions and updated the manuscript to address every concern raised. Below is a summary of the primary concerns raised by each reviewer and the outcomes of our rebuttal:

**Reviewer dQeU (Rating: 4 → 6, Nov 21)**

[revision of reviewer dQeU in Link: https://openreview.net/revisions?id=r7tTc7rgPf](https://openreview.net/revisions?id=r7tTc7rgPf)

**● Key Concerns:** Requested clearer theoretical grounding (update frequency _T_ and _τ_), and suggested for datasets beyond U.S. cities.
**● Response & Outcome:** We clarified the rationale behind selecting _T_ and _τ_, explaining how different choices address various types of concept drift (Sec. 5.3). We also added a new large-scale crime dataset (Toronto, Canada), conducted full experiments, and expanded the complexity analysis (Sec. 5.1). The reviewer found the responses satisfactory and raised the score to 6.

-----

**Reviewer pemE (Rating: 6)**
**● Key Concerns:** Sought a clearer articulation of the method’s contributions, justification of using a partially frozen LLM, and more evidence of PF-LLM effectiveness.
**● Response & Outcome:** We strengthened the discussion of contributions and added comparisons to related work in Sec. 2, explaining that the partially frozen GPT-2 (Small) is a sufficient choice. We also expanded the ablation studies in Sec. 5.2. The reviewer has not yet responded further.

----

**Reviewer GDKr (Rating: 2 → 6, Nov 17)**

[revision of reviewer GDKr in Link: https://openreview.net/revisions?id=LlLDIOkTau](https://openreview.net/revisions?id=LlLDIOkTau)

**● Key Concerns:** Asked for the design motivations of each module, distinctions from prior work, handling of data sparsity, consideration of stronger LLMs, and a more complete ethics discussion.
**● Response & Outcome:** We provided detailed explanations of the design motivations and distinctions of every module, and clarified our contributions. We reinforced how our method addresses data sparsity in Sec. 4, noted that exploring stronger LLMs is not our goal, and expanded the Ethics Statement in Sec. 8. The reviewer confirmed that all concerns were resolved and updated the score to 6.

----

**Reviewer 9AaK (Rating: 8, maintain)**
**● Key Concerns:** Requested discussion on computational complexity, how each module addresses sparsity, and the definition of online learning in this context.
**● Response & Outcome:** We elaborated on scalability, clarified how all three modules mitigate sparsity, and explained the definition of online learning in spatio-temporal forecasting. The reviewer expressed satisfaction and maintained the score of 8.

----

**Reviewer 3QXX (Rating: 4 → 6, Nov 24)**

[revision of reviewer 3QXX in Link: https://openreview.net/revisions?id=1LsYH980lK](https://openreview.net/revisions?id=1LsYH980lK)

**● Key Concerns:** Sought clarification on sparsity handling, inclusion of a True Zero Rate metric, and ablation results across all datasets.
**● Response & Outcome:** We expanded Sec. 4 to explain how each module addresses sparsity, added the True Zero Rate evaluation in Sec. 5.1, and reported full-dataset ablation results in Sec. 5.2. The reviewer found the updates sufficient and raised the score to 6.

----

**We deeply appreciate the AC’s and reviewers’ hard work, especially under the special circumstances of this cycle. We trust AC with expertise and leadership in the field, will make a fair and well-informed judgment**. Thank you again for your constructive feedback and for engaging with our submission.

Best regards,

The Authors of Submission 4693

---

### Meta-Review · Area_Chair_ptvA · 2026-01-02

**Summary:**

The reviewers' concerns centered on the need for stronger theoretical justification of the online learning mechanism and update frequencies; clarity on the novelty and distinct contributions beyond simply combining existing components like hypergraphs and LLMs; experimental completeness regarding dataset diversity, comprehensive ablation studies, and inclusion of critical metrics like True Zero Rate; technical details on scalability and how the model handles data sparsity; and a more thorough discussion of the ethical implications of predictive policing. These points were raised constructively to strengthen the paper's rigor, impact, and clarity.

**Reviewer Concerns:**

The authors' rebuttal has effectively addressed the vast majority of the concerns raised. They provided theoretical grounding through FFT analysis to justify update schedules, clearly articulated the novel architectural innovations of their hierarchical hypergraph design, and significantly expanded their experimental section. This included adding a new international dataset (Toronto), conducting full ablation studies across all datasets (including removing the PF-LLM), and incorporating the True Zero Rate metric. They also expanded the ethics discussion and clarified the definition of online learning in their context. All reviewers who engaged with the rebuttal confirmed their concerns were resolved.

**Reviewer Scores:**

The rebuttal process resulted in clear, positive momentum. Three reviewers (dQeU, GDKr, and 3QXX) explicitly increased their scores from below the acceptance threshold to a solid 6 after their concerns were fully addressed. Reviewer 9AaK, who initially gave a high score of 8, maintained this score after expressing satisfaction with the clarifications. While pemE did not provide a follow-up comment, the authors offered a substantial response to their queries, including new ablation results. It is reasonable to estimate that pemE would likely maintain or potentially even increase their initial score of 6, leading to a final set of scores that are consistently positive and supportive of acceptance.

---

### Decision · Program_Chairs · 2026-01-26

Accept (Poster)